# On Representing Linear Programs by Graph Neural Networks

**Ziang Chen** [*]
Department of Mathematics, Duke University
Durham, NC 27708
`ziang@math.duke.edu`

**Jialin Liu** [†]
Damo Academy, Alibaba US
Bellevue, WA 98004
`jialin.liu@alibaba-inc.com`

**Xinshang Wang**
Damo Academy, Alibaba US
Antai College of Economics and Management, Shanghai Jiao Tong University
Bellevue, WA 98004
`xinshang.w@alibaba-inc.com`

**Jianfeng Lu**
Departments of Mathematics, Physics,
and Chemistry, Duke University
Durham, NC 27708
`jianfeng@math.duke.edu`

**Wotao Yin**
Damo Academy, Alibaba US
Bellevue, WA 98004
`wotao.yin@alibaba-inc.com`

## Abstract

Learning to optimize is a rapidly growing area that aims to solve optimization problems or improve existing optimization algorithms using machine learning (ML). In particular, the graph neural network (GNN) is considered a suitable ML model for optimization problems whose variables and constraints are permutation–invariant, for example, the linear program (LP). While the literature has reported encouraging numerical results, this paper establishes the theoretical foundation of applying GNNs to solving LPs. Given any size limit of LPs, we construct a GNN that maps different LPs to different outputs. We show that properly built GNNs can reliably predict feasibility, boundedness, and an optimal solution for each LP in a broad class. Our proofs are based upon the recently–discovered connections between the Weisfeiler–Lehman isomorphism test and the GNN. To validate our results, we train a simple GNN and present its accuracy in mapping LPs to their feasibilities and solutions.

## 1 Introduction

Applying machine learning (ML) techniques to accelerate optimization, also known as *Learning to Optimize (L2O)*, is attracting increasing attention. It has been reported that L2O shows great potentials on both continuous optimization (Monga et al., 2021; Chen et al., 2021; Amos, 2022) and combinatorial optimization (Bengio et al., 2021; Mazyavkina et al., 2021). Many of the L2O works train a parameterized model that takes the optimization problem as input and outputs information useful to classic algorithms, such as a good initial solution and branching decisions (Nair et al., 2020), and some even directly generate an approximate optimal solution (Gregor & LeCun, 2010).

In these works, one is building an ML model to approximate the mapping from an explicit optimization instance either to its key properties or directly to its solution. The ability to achieve accurate approximation is called the representation power or expressive power of the model. When the approximation is accurate, the model can solve the problem or provide useful information to guide an optimization algorithm. This paper tries to address a fundamental but open theoretical problem for

---

[*]A major part of the work of Z. Chen was completed during his internship at Alibaba US DAMO Academy.
[†]Corresponding author.

linear programming (LP):

> *Which neural network can represent LP and predict its key properties and solution?* (P0)

To clarify, by solution we mean the optimal solution. Let us also remark that this question is not only of theoretical interest. Although currently neural network models may not be powerful enough to replace those mathematical-grounded LP solvers and obtain an exact LP solution, they are still useful in helping LP solvers from several perspectives, including warm-start and configuration. It requires that neural networks have sufficient power to recognize key characteristics of LPs. Some very recent papers (Deka & Misra, 2019; Pan et al., 2020; Chen et al., 2022) on DC optimal power flow (DC-OPF), an important type of LP, experimentally show the possibility of fast approximating LP solutions with deep neural networks. Practitioners may initialize an LP solver with those approximated solutions. We hope the answer to (P0) paves the way toward answering this question for other optimization types.

**Linear Programming (LP).** LP is an important type of optimization problem with a wide range of applications, such as scheduling (Hanssmann & Hess, 1960), signal processing (Candes & Tao, 2005), machine learning (Dedieu et al., 2022), etc. A general LP problem is defined as:

$$\min_{x \in \mathbb{R}^n} \quad c^\top x, \quad \text{s.t.} \ Ax \circ b, \ l \le x \le u, \tag{1.1}$$

where $A \in \mathbb{R}^{m \times n}$, $c \in \mathbb{R}^n$, $b \in \mathbb{R}^m$, $l \in (\mathbb{R} \cup \{-\infty\})^n$, $u \in (\mathbb{R} \cup \{+\infty\})^n$, and $\circ \in \{\le, =, \ge\}^m$. Any LP problems must follow one of the following three cases (Bertsimas & Tsitsiklis):

- *Infeasible.* The feasible set $\mathcal{X}_F := \{x \in \mathbb{R}^n : Ax \circ b, \ l \le x \le u\}$ is empty. In another word, there is no point in $\mathbb{R}^n$ that satisfies the constraints in LP (1.1).
- *Unbounded.* The feasible set is non-empty, but the objective value can be arbitrarily good, i.e., unbounded from below. For any $R > 0$, there exists an $x \in \mathcal{X}_F$ such that $c^\top x < -R$.
- *Feasible and bounded.* There exists $x^* \in \mathcal{X}_F$ such that $c^\top x^* \le c^\top x$ for all $x \in \mathcal{X}_F$. Such $x^*$ is named as an optimal solution, and $c^\top x^*$ is the optimal objective value.

Thus, considering (P0), an ideal ML model is expected to be able to predict the three key characteristics of LP: *feasibility, boundedness, and one of its optimal solutions* (if exists), by taking the LP features $(A, b, c, l, u, \circ)$ as input. Actually, such input has a strong mathematical structure. If we swap the positions of the $i, j$-th variable in (1.1), elements in vectors $b, c, l, u, \circ$ and columns of matrix $A$ will be reordered. The reordered features $(\hat{A}, b, \hat{c}, \hat{l}, \hat{u}, \hat{\circ})$ actually represent an exactly equivalent LP problem with the original one $(A, b, c, l, u, \circ)$. Such property is named as *permutation invariance*. If we do not explicitly restrict ML models with a permutation invariant structure, the models may overfit to the variable/constraint orders of instances in the training set. Motivated by this point, we adopt *Graph Neural Networks (GNNs)* that are permutation invariant naturally.

**GNN in L2O.** GNN is a type of neural networks defined on graphs and widely applied in many areas, for example, recommender systems, traffic, chemistry, etc (Wu et al., 2020; Zhou et al., 2020). Accelerating optimization solvers with GNNs attracts rising interest recently (Peng et al., 2021; Cappart et al., 2021). Many graph-related optimization problems, like minimum vertex cover, traveling salesman, vehicle routing, can be represented and solved approximately with GNNs due to their problem structures (Khalil et al., 2017; Kool et al., 2019; Joshi et al., 2019; Drori et al., 2020). Besides that, one may solve a general LP or mixed-integer linear programming (MILP) with the help of GNNs. Gasse et al. (2019) proposed to represent an MILP with a bipartite graph and apply a GNN on this graph to guide an MILP solver. Ding et al. (2020) proposed a tripartite graph to represent MILP. Since that, many approaches have been proposed to guide MILP or LP solvers with GNNs (Nair et al., 2020; Gupta et al., 2020; 2022; Shen et al., 2021; Khalil et al., 2022; Liu et al., 2022; Paulus et al., 2022; Qu et al., 2022; Li et al., 2022). Although encouraging empirical results have been observed, theoretical foundations are still lack for this approach. Specifying (P0), we ask:

> *Are there GNNs that can predict the feasibility, boundedness and an optimal solution of LP?* (P1)

**Related works and contributions.** To answer (P1), one needs the theory of *separation power* and *representation power*. Separation power of a neural network (NN) means its ability to distinguish two different inputs. In our settings, a NN with strong separation power means that it can outputs different results when it is applied on any two different LPs. Representation power of NN means its ability to approximate functions of interest. The theory of representation power is established upon the separation power. Only functions with strong enough separation power may possess strong

representation power. The power of GNN has been studied in the literature (see Sato (2020); Jegelka (2022); Li & Leskovec (2022) for comprehensive surveys), and some theoretical efforts have been made to represent some graph-related optimization problems with GNN (Sato et al., 2019; Loukas, 2020). However, there are still gaps to answer question (P1) since the relationships between characteristics of LP and properties of graphs are not well established. Our contributions are listed below:

- (Separation Power). In the literature, it has been shown that the separation power of GNN is equal to the WL test (Xu et al., 2019; Azizian & Lelarge, 2021; Geerts & Reutter, 2022). However, there exist many pairs of LPs that cannot be distinguished by the WL test. We show that those puzzling LP pairs share the same feasibility, boundedness, and even an optimal solution if exists. Thus, GNN has strong enough separation power.
- (Representation Power). To the best of our knowledge, we established the first complete proof that GNN can universally represent a broad class of LPs. More precisely, we prove that, there exist GNNs that can be arbitrarily close to the following three mappings: LP → feasibility, LP → optimal objective value ($-\infty$ if unbounded and $\infty$ if infeasible), and LP → an optimal solution (if exists), although they are not continuous functions and cannot be covered by the literature (Keriven & Peyré, 2019; Chen et al., 2019; Maron et al., 2019a;b; Keriven et al., 2021).
- (Experimental Validation). We design and conduct experiments that demonstrate the power of GNN on representing LP.

The rest of this paper is organized as follows. In Section 2, we provide preliminaries, including related notions, definitions and concepts. In Section 3, we present our main theoretical results. The sketches of proofs are provided in Section4. We validate our results with numerical experiments in Section 5 and we conclude this paper with Section 6.

## 2 PRELIMINARIES

In this section, we present concepts and definitions that will be used throughout this paper. We first describe how to represent an LP with a weighted bipartite graph, then we define GNN on those LP-induced graphs, and finally we further clarify question (P1) with strict mathematical definitions.

### 2.1 LP REPRESENTED AS WEIGHTED BIPARTITE GRAPH

Before representing LPs with graphs, we first define the graph that we will adopt in this paper: *weighted bipartite graph*. A weighted bipartite graph $G = (V \cup W, E)$ consists of a vertex set $V \cup W$ that are divided into two groups $V$ and $W$ with $V \cap W = \emptyset$, and a collection $E$ of weighted edges, where each edge connects exactly one vertex in $V$ and one vertex in $W$. Note that there is no edge connecting vertices in the same vertex group. $E$ can also be viewed as a function $E : V \times W \to \mathbb{R}$. We use $\mathcal{G}_{m,n}$ to denote the collection of all weighted bipartite graphs $G = (V \cup W, E)$ with $|V| = m$ and $|W| = n$. We always write $V = \{v_1, v_2, \ldots, v_m\}$, $W = \{w_1, w_2, \ldots, w_n\}$, and $E_{i,j} = E(v_i, w_j)$, for $i \in \{1, 2, \ldots, m\}$, $j \in \{1, 2, \ldots, n\}$.

One can equip each vertex with a feature vector. Throughout this paper, we denote $h_i^V \in \mathcal{H}^V$ as the feature vector of vertex $v_i \in V$ and denote $h_j^W \in \mathcal{H}^W$ as the feature vector of vertex $w_j \in W$, where $\mathcal{H}^V, \mathcal{H}^W$ are feature spaces. Then we define $\mathcal{H}_m^V := (\mathcal{H}^V)^m, \mathcal{H}_n^W := (\mathcal{H}^W)^n$ and concatenate all the vertex features together as $H = (h_1^V, h_2^V, \ldots, h_m^V, h_1^W, h_2^W, \ldots, h_n^W) \in \mathcal{H}_m^V \times \mathcal{H}_n^W$. Finally, a weighted bipartite graph with vertex features is defined as a tuple $(G, H) \in \mathcal{G}_{m,n} \times \mathcal{H}_m^V \times \mathcal{H}_n^W$.

With the concepts described above, one can represent an LP (1.1) as a bipartite graph (Gasse et al., 2019): Each vertex in $W$ represents a variable in LP and each vertex in $V$ represents a constraint. The graph topology is defined with the matrix $A$ in the linear constraint. More specifically, let us set

- Vertex $v_i$ represents the $i$-th constraint in $Ax \circ b$, and vertex $w_j$ represents the $j$-th variable $x_j$.
- Information of constraints is involved in the feature of $v_i$: $h_i^V = (b_i, \circ_i)$.
- The space of constraint features is defined as $\mathcal{H}^V := \mathbb{R} \times \{\leq, =, \geq\}$.
- Information of variables is involved in the feature of $w_i$: $h_j^W = (c_j, l_j, u_j)$.
- The space of variable features is defined as $\mathcal{H}^W := \mathbb{R} \times (\mathbb{R} \cup \{-\infty\}) \times (\mathbb{R} \cup \{+\infty\})$.
- The edge connecting $v_i$ and $w_j$ has weight $E_{i,j} = A_{i,j}$.

Then an LP is represented as a graph $(G, H) \in \mathcal{G}_{m,n} \times \mathcal{H}_m^V \times \mathcal{H}_n^W$. In the rest of this paper, we coin such graphs as *LP-induced Graphs* or *LP-Graphs* for simplicity. We present an LP instance and its corresponding LP-graph in Figure 1.

$$\min_{x \in \mathbb{R}^2} \quad x_1 + 2x_2,$$
$$\text{s.t.} \quad x_1 + 2x_2 \geq 1,$$
$$2x_1 + x_2 = 2,$$
$$x_1 \geq 0, \ x_2 \geq -1.$$

$h_1^V = (1, \geq)$   $v_1$  —1—  $w_1$   $h_1^W = (1, 0, +\infty)$

$h_2^V = (2, =)$   $v_2$  —1—  $w_2$   $h_2^W = (2, -1, +\infty)$

Figure 1: An example of LP-graph

## 2.2 Graph neural networks for LP

The GNNs in this paper always take an LP-Graph as input and the output has two cases:

- The output is a single real number. In this case, GNN is a function $\mathcal{G}_{m,n} \times \mathcal{H}_m^V \times \mathcal{H}_n^W \to \mathbb{R}$ and usually used to predict the properties of the whole graph.

- Each vertex in $W$ has an output. Consequently, GNN is a function $\mathcal{G}_{m,n} \times \mathcal{H}_m^V \times \mathcal{H}_n^W \to \mathbb{R}^n$. Since $W$ represents variables in LP, GNN is used to predict properties of each variable in this case.

Now we define the GNN structure precisely. First we encode the input features into the embedding space with learnable functions $f_{\text{in}}^V : \mathcal{H}^V \to \mathbb{R}^{d_0}$ and $f_{\text{in}}^W : \mathcal{H}^W \to \mathbb{R}^{d_0}$:

$$h_i^{0,V} = f_{\text{in}}^V(h_i^V), \ \ h_j^{0,W} = f_{\text{in}}^W(h_j^W), \ \ i = 1, 2, \ldots, m, \ j = 1, 2, \ldots, n, \tag{2.1}$$

where $h_i^{0,V}, h_j^{0,W} \in \mathbb{R}^{d_0}$ are initial embedded vertex features and $d_0$ is their dimension. Then we choose learnable functions $f_l^V, f_l^W : \mathbb{R}^{d_{l-1}} \to \mathbb{R}^{d_l}$ and $g_l^V, g_l^W : \mathbb{R}^{d_{l-1}} \times \mathbb{R}^{d_l} \to \mathbb{R}^{d_l}$ and update the hidden states with[1]:

$$h_i^{l,V} = g_l^V\left(h_i^{l-1,V}, \sum_{j=1}^n E_{i,j} f_l^W(h_j^{l-1,W})\right), \quad i = 1, 2, \ldots, m, \tag{2.2}$$

$$h_j^{l,W} = g_l^W\left(h_j^{l-1,W}, \sum_{i=1}^m E_{i,j} f_l^V(h_i^{l-1,V})\right), \quad j = 1, 2, \ldots, n, \tag{2.3}$$

where $h_i^{l,V}, h_j^{l,W} \in \mathbb{R}^{d_l}$ are vertex features at layer $l$ $(1 \leq l \leq L)$ and their dimensions are $d_1, \cdots, d_L$ respectively. The output layer of the single-output GNN is defined with a learnable function $f_{\text{out}} : \mathbb{R}^{d_L} \times \mathbb{R}^{d_L} \to \mathbb{R}$:

$$y_{\text{out}} = f_{\text{out}}\left(\sum_{i=1}^m h_i^{L,V}, \sum_{j=1}^n h_j^{L,W}\right). \tag{2.4}$$

The output of the vertex-output GNN is defined with $f_{\text{out}}^W : \mathbb{R}^{d_L} \times \mathbb{R}^{d_L} \times \mathbb{R}^{d_L} \to \mathbb{R}$:

$$y_{\text{out}}(w_j) = f_{\text{out}}^W\left(\sum_{i=1}^m h_i^{L,V}, \sum_{j=1}^n h_j^{L,W}, h_j^{L,W}\right), \quad j = 1, 2, \cdots, n. \tag{2.5}$$

We denote collections of single-output GNNs and vertex-output GNNs with $\mathcal{F}_{\text{GNN}}$ and $\mathcal{F}_{\text{GNN}}^W$, respectively:

$$\mathcal{F}_{\text{GNN}} = \{F : \mathcal{G}_{m,n} \times \mathcal{H}_m^V \times \mathcal{H}_n^W \to \mathbb{R} \mid F \text{ yields } (2.1), (2.2), (2.3), (2.4)\},$$
$$\mathcal{F}_{\text{GNN}}^W = \{F : \mathcal{G}_{m,n} \times \mathcal{H}_m^V \times \mathcal{H}_n^W \to \mathbb{R}^n \mid F \text{ yields } (2.1), (2.2), (2.3), (2.5)\}. \tag{2.6}$$

In practice, all the learnable functions in GNN $f_{\text{in}}^V, f_{\text{in}}^W, f_{\text{out}}, f_{\text{out}}^W, \{f_l^V, f_l^W, g_l^V, g_l^W\}_{l=0}^L$ are usually parameterized with multi-linear perceptrons (MLPs). In our theoretical analysis, we assume for simplicity that those functions may take all continuous functions on given domains, following the settings in Azizian & Lelarge (2021, Section C.1). Thanks to the universal approximation properties of MLP (Hornik et al., 1989; Cybenko, 1989), one can extend our theoretical results by taking those learnable functions as large enough MLPs.

## 2.3 Revisiting question (P1)

With the definitions of LP-Graph and GNN above, we revisit the question (P1) and provide its precise mathematical description here. First we define three mappings that respectively describe the feasibility, optimal objective value and an optimal solution of an LP (if exists).

---

[1] Note that the update rules in (2.2) and (2.3) follow a message-passing way, where each vertex only collects information from its neighbors. Since $E_{i,j} = 0$ if there is no connection between vertices $v_i$ and $w_j$, the sum operator in (2.2) can be rewritten as $\sum_{j \in \mathcal{N}(v_i)}$, where $\mathcal{N}(v_i)$ denotes the set of neighbors of vertex $v_i$.

**Feasibility mapping**    The feasibility mapping is a classification function
$$\Phi_{\text{feas}} : \mathcal{G}_{m,n} \times \mathcal{H}_m^V \times \mathcal{H}_n^W \to \{0, 1\}, \tag{2.7}$$
where $\Phi_{\text{feas}}(G, H) = 1$ if the LP associated with $(G, H)$ is feasible and $\Phi_{\text{feas}}(G, H) = 0$ otherwise.

**Optimal objective value mapping**    Denote
$$\Phi_{\text{obj}} : \mathcal{G}_{m,n} \times \mathcal{H}_m^V \times \mathcal{H}_n^W \to \mathbb{R} \cup \{\infty, -\infty\}, \tag{2.8}$$
as the optimal objective value mapping, i.e., for any $(G, H) \in \mathcal{G}_{m,n} \times \mathcal{H}_m^V \times \mathcal{H}_n^W$, $\Phi_{\text{obj}}(G, H)$ is the optimal objective value of the LP problem associated with $(G, H)$.

**Remark 2.1.** *The optimal objective value of a LP problem can be a real number or $\infty / - \infty$. The "$\infty$" case corresponds to infeasible problems, while the "$-\infty$" case consists of LP problems whose objective function is unbounded from below in the feasible region. The preimage of all finite real numbers under $\Phi_{obj}$, $\Phi_{obj}^{-1}(\mathbb{R})$, actually describes all LPs with finite optimal objective value.*

**Remark 2.2.** *In the case that a LP problem has finite optimal objective value, it is possible that the problem admits multiple optimal solutions. However, the optimal solution with the smallest $\ell_2$-norm must be unique. In fact, if $x \neq x'$ are two different solutions with $\|x\| = \|x'\|$, where $\|\cdot\|$ denotes the $\ell_2$-norm throughout this paper. Then $\frac{1}{2}(x + x')$ is also an optimal solution due to the convexity of LPs, and it holds that $\|\frac{1}{2}(x + x')\|^2 < \frac{1}{2}\|x\|^2 + \frac{1}{2}\|x'\|^2 = \|x\|^2 = \|x'\|^2$, i.e., $\|\frac{1}{2}(x + x')\| < \|x\| = \|x'\|$, where the inequality is strict since $x \neq x'$. Therefore, $x$ and $x'$ cannot be optimal solutions with the smallest $\ell_2$-norm.*

**Optimal solution mapping**    For any $(G, H) \in \Phi_{\text{obj}}^{-1}(\mathbb{R})$, we have remarked before that the LP problem associated with $(G, H)$ has a unique optimal solution with the smallest $\ell_2$-norm. Let
$$\Phi_{\text{solu}} : \Phi_{\text{obj}}^{-1}(\mathbb{R}) \to \mathbb{R}^n, \tag{2.9}$$
be the mapping that maps $(G, H) \in \Phi_{\text{obj}}^{-1}(\mathbb{R})$ to the optimal solution with the smallest $\ell_2$-norm.

**Invariance and Equivariance**    We denote $S_m, S_n$ as the group consisting of all permutations on vertex groups $V, W$ respectively. In another word, $S_m$ involves all permutations on the constraints of LP and $S_n$ involves all permutations on the variables. In this paper, we say a function $F : \mathcal{G}_{m,n} \times \mathcal{H}_m^V \times \mathcal{H}_n^W \to \mathbb{R}$ is invariant if it satisfies
$$F(G, H) = F((\sigma_V, \sigma_W) * (G, H)), \quad \forall \sigma_V \in S_m, \sigma_W \in S_n,$$
and a function $F_W : \mathcal{G}_{m,n} \times \mathcal{H}_m^V \times \mathcal{H}_n^W \to \mathbb{R}^n$ is equivariant if it satisfies
$$\sigma_W(F_W(G, H)) = F_W((\sigma_V, \sigma_W) * (G, H))), \quad \forall \sigma_V \in S_m, \sigma_W \in S_n,$$
where $(\sigma_V, \sigma_W) * (G, H)$ is the permuted graph obtained from reordering indices in $(G, H)$ using $(\sigma_V, \sigma_W)$, which is the group action of $S_m \times S_n$ on $\mathcal{G}_{m,n} \times \mathcal{H}_m^V \times \mathcal{H}_n^W$. One can check that $\Phi_{\text{feas}}$, $\Phi_{\text{obj}}$, and any $F \in \mathcal{F}_{\text{GNN}}$ are invariant, and that $\Phi_{\text{solu}}$ and any $F_W \in \mathcal{F}_{\text{GNN}}^W$ are equivariant.

Question (P1) actually asks: Does there exist $F \in \mathcal{F}_{\text{GNN}}$ that well approximates $\Phi_{\text{feas}}$ or $\Phi_{\text{obj}}$? And does there exist $F_W \in \mathcal{F}_{\text{GNN}}^W$ that well approximates $\Phi_{\text{solu}}$?

## 3    MAIN RESULTS

This section presents our main theorems that answer question (P1). As we state in the introduction, representation power is built upon separation power in our paper. We first present with the following theorem that GNN has strong enough separation power to represent LP.

**Theorem 3.1.** *Given any two LP instances $(G, H), (\hat{G}, \hat{H}) \in \mathcal{G}_{m,n} \times \mathcal{H}_m^V \times \mathcal{H}_n^W$, if $F(G, H) = F(\hat{G}, \hat{H})$ for all $F \in \mathcal{F}_{GNN}$, then they share some common characteristics:*

*(i) Both LP problems are feasible or both are infeasible, i.e., $\Phi_{feas}(G, H) = \Phi_{feas}(\hat{G}, \hat{H})$.*

*(ii) The two LP problems have the same optimal objective value, i.e., $\Phi_{obj}(G, H) = \Phi_{obj}(\hat{G}, \hat{H})$.*

*(iii) If both problems are feasible and bounded, they have the same optimal solution with the smallest $\ell_2$-norm up to a permutation, i.e., $\Phi_{solu}(G, H) = \sigma_W(\Phi_{solu}(\hat{G}, \hat{H}))$ for some $\sigma_W \in S_n$.*

*Furthermore, if $F_W(G, H) = F_W(\hat{G}, \hat{H}), \forall F_W \in \mathcal{F}_{GNN}^W$, then (iii) holds without taking permutations, i.e., $\Phi_{solu}(G, H) = \Phi_{solu}(\hat{G}, \hat{H})$.*

This theorem demonstrates that the function spaces $\mathcal{F}_{\text{GNN}}$ and $\mathcal{F}_{\text{GNN}}^W$ are rich enough to distinguish the characteristics of LP. Given two LP instances $(G, H), (\hat{G}, \hat{H})$, as long as their feasibility or boundedness are different, there must exist $F \in \mathcal{F}_{\text{GNN}}$ that can distinguish them: $F(G, H) \neq F(\hat{G}, \hat{H})$. Moreover, as long as their optimal solutions with the smallest $\ell_2$-norm are different, there must exist $F_W \in \mathcal{F}_{\text{GNN}}^W$ that can distinguish them: $F_W(G, H) \neq F_W(\hat{G}, \hat{H})$. With Theorem 3.1 served as a foundation, we can prove that GNN can approximate the three mappings $\Phi_{\text{feas}}$, $\Phi_{\text{obj}}$ and $\Phi_{\text{solu}}$ to arbitrary precision. Before presenting those results, we first define some concepts of the space $\mathcal{G}_{m,n} \times \mathcal{H}_m^V \times \mathcal{H}_n^W$.

**Topology and measure** Throughout this paper, we consider $\mathcal{G}_{m,n} \times \mathcal{H}_m^V \times \mathcal{H}_n^W$, where $\mathcal{H}^V = \mathbb{R} \times \{\leq, =, \geq\}$ and $\mathcal{H}^W = \mathbb{R} \times (\mathbb{R} \cup \{-\infty\}) \times (\mathbb{R} \cup \{+\infty\})$, as a topology space with product topology and a measurable space with product measure. It's enough to define the topology and measure of each part separately. Since each graph in this paper (without vertex features) can be represented with matrix $A \in \mathbb{R}^{m \times n}$, the graph space $\mathcal{G}_{m,n}$ is isomorphic to the Euclidean space $\mathbb{R}^{m \times n}$: $\mathcal{G}_{m,n} \cong \mathbb{R}^{m \times n}$ and we equip $\mathcal{G}_{m,n}$ with the standard Euclidean topology and the standard Lebesgue measure. The real spaces $\mathbb{R}$ in $\mathcal{H}^W$ and $\mathcal{H}^V$ are also equipped with the standard Euclidean topology and Lebesgue measure. All the discrete spaces $\{\leq, =, \geq\}$, $\{-\infty\}$, and $\{+\infty\}$ have the discrete topology, and all unions are disjoint unions. We equip those spaces with a discrete measure $\mu(S) = |S|$, where $|S|$ is the number of elements in a finite set $S$. This finishes the whole definition and we denote $\text{Meas}(\cdot)$ as the measure on $\mathcal{G}_{m,n} \times \mathcal{H}_m^V \times \mathcal{H}_n^W$.

**Theorem 3.2.** *Given any measurable $X \subset \mathcal{G}_{m,n} \times \mathcal{H}_m^V \times \mathcal{H}_n^W$ with finite measure, for any $\epsilon > 0$, there exists some $F \in \mathcal{F}_{GNN}$, such that*

$$\text{Meas}\left(\left\{(G, H) \in X : \mathbb{I}_{F(G,H)>1/2} \neq \Phi_{feas}(G, H)\right\}\right) < \epsilon,$$

*where $\mathbb{I}_.$ is the indicator function, i.e., $\mathbb{I}_{F(G,H)>1/2} = 1$ if $F(G, H) > 1/2$ and $\mathbb{I}_{F(G,H)>1/2} = 0$ otherwise.*

This theorem shows that GNN is a good classifier for LP instances in $X$ as long as $X$ has finite measure. If we use $F(G, H) > 1/2$ as the criteria to predict the feasibility, the classification error rate is controlled by $\epsilon/\text{Meas}(X)$, where $\epsilon$ can be arbitrarily small. Furthermore, we show that GNN can perfectly fit any dataset with finite samples, which is presented in the following corollary.

**Corollary 3.3.** *For any $\mathcal{D} \subset \mathcal{G}_{m,n} \times \mathcal{H}_m^V \times \mathcal{H}_n^W$ with finite instances, there exists $F \in \mathcal{F}_{GNN}$ that*

$$\mathbb{I}_{F(G,H)>1/2} = \Phi_{feas}(G, H), \quad \forall (G, H) \in \mathcal{D}.$$

Besides the feasibility, GNN can also approximate $\phi_{\text{obj}}$ and $\phi_{\text{solu}}$.

**Theorem 3.4.** *Given any measurable $X \subset \mathcal{G}_{m,n} \times \mathcal{H}_m^V \times \mathcal{H}_n^W$ with finite measure, for any $\epsilon > 0$, there exists $F_1 \in \mathcal{F}_{GNN}$ such that*

$$\text{Meas}\left(\left\{(G, H) \in X : \mathbb{I}_{F_1(G,H)>1/2} \neq \mathbb{I}_{\Phi_{obj}(G,H)\in\mathbb{R}}\right\}\right) < \epsilon, \tag{3.1}$$

*and for any $\epsilon, \delta > 0$, there exists $F_2 \in \mathcal{F}_{GNN}$ such that*

$$\text{Meas}\left(\left\{(G, H) \in X \cap \Phi_{obj}^{-1}(\mathbb{R}) : |F_2(G, H) - \Phi_{obj}(G, H)| > \delta\right\}\right) < \epsilon. \tag{3.2}$$

Recall the definition of $\Phi_{\text{obj}}$ in (2.8) that it can take $\{\pm\infty\}$ as its value. Thus, $\Phi_{\text{obj}}(G, H) \in \mathbb{R}$ means the LP corresponding to $(G, H)$ is feasible and bounded with a finite optimal objective value, and inequality (3.1) illustrates that GNN can identify those feasible and bounded LPs among the whole set $X$, up to a given precision $\epsilon$. Inequality (3.2) shows that GNN can also approximate the optimal value. The measure of the set of LP instances of which the optimal value cannot be approximated with $\delta$-precision is controlled by $\epsilon$. The following corollary gives the results on dataset with finite instances.

**Corollary 3.5.** *For any $\mathcal{D} \subset \mathcal{G}_{m,n} \times \mathcal{H}_m^V \times \mathcal{H}_n^W$ with finite instances, there exists $F_1 \in \mathcal{F}_{GNN}$ such that*

$$\mathbb{I}_{F_1(G,H)>1/2} = \mathbb{I}_{\Phi_{obj}(G,H)\in\mathbb{R}}, \quad \forall (G, H) \in \mathcal{D},$$

*and for any $\delta > 0$, there exists $F_2 \in \mathcal{F}_{GNN}$, such that*

$$|F_2(G, H) - \Phi_{obj}(G, H)| < \delta, \quad \forall (G, H) \in \mathcal{D} \cap \Phi_{obj}^{-1}(\mathbb{R}).$$

Finally, we show that GNN is able to represent the optimal solution mapping $\Phi_{\text{solu}}$.

**Theorem 3.6.** *Given any measurable $X \subset \Phi_{obj}^{-1}(\mathbb{R}) \subset \mathcal{G}_{m,n} \times \mathcal{H}_m^V \times \mathcal{H}_n^W$ with finite measure, for any $\epsilon, \delta > 0$, there exists some $F_W \in \mathcal{F}_{GNN}^W$, such that*

$$\text{Meas}\left(\{(G, H) \in X : \|F(G, H) - \Phi_{solu}(G, H)\| > \delta\}\right) < \epsilon.$$

**Corollary 3.7.** *Given any $\mathcal{D} \subset \Phi_{obj}^{-1}(\mathbb{R}) \subset \mathcal{G}_{m,n} \times \mathcal{H}_m^V \times \mathcal{H}_n^W$ with finite instances, for any $\delta > 0$, there exists $F_W \in \mathcal{F}_{GNN}^W$, such that*

$$\|F(G, H) - \Phi_{solu}(G, H)\| < \delta, \quad \forall (G, H) \in \mathcal{D}.$$

## 4 SKETCH OF PROOF

In this section, we will present a sketch of our proof lines and provide examples to show the intuitions. The full proof lines are presented in the appendix.

**Separation power** The separation power measures a neural network with whether it generates different outcomes given different inputs, which serves as a foundation of the representation power. The separation power of GNNs is closely related to the Weisfeiler-Lehman (WL) test (Weisfeiler & Leman, 1968), a classical algorithm to identify whether two given graphs are isomorphic. To apply the WL test on LP-graphs, we describe a modified WL test in Algorithm 1, which is slightly different from the standard WL test.

---

**Algorithm 1** The WL test for LP-Graphs[2](denoted by $\text{WL}_{\text{LP}}$)

---

**Require:** A graph instance $(G, H) \in \mathcal{G}_{m,n} \times \mathcal{H}_m^V \times \mathcal{H}_n^W$ and iteration limit $L > 0$.
1: Initialize with $C_i^{0,V} = \text{HASH}_{0,V}(h_i^V)$, $C_j^{0,W} = \text{HASH}_{0,W}(h_j^W)$.
2: **for** $l = 1, 2, \cdots, L$ **do**
3: $\quad C_i^{l,V} = \text{HASH}_{l,V}\left(C_i^{l-1,V}, \sum_{j=1}^n E_{i,j}\text{HASH}_{l,W}'\left(C_j^{l-1,W}\right)\right).$
4: $\quad C_j^{l,W} = \text{HASH}_{l,W}\left(C_j^{l-1,W}, \sum_{i=1}^m E_{i,j}\text{HASH}_{l,V}'\left(C_i^{l-1,V}\right)\right).$
5: **end for**
6: **return** The multisets containing all colors $\{\{C_i^{L,V}\}\}_{l=0}^m, \{\{C_j^{L,W}\}\}_{j=0}^n.$

---

We denote Algorithm 1 by $\text{WL}_{\text{LP}}(\cdot)$, and we say that two LP-graphs $(G, H), (\hat{G}, \hat{H})$ *can be distinguished by Algorithm 1* if and only if there exist a positive integer $L$ and injective hash functions $\{\text{HASH}_{l,V}, \text{HASH}_{l,W}\}_{l=0}^L \cup \{\text{HASH}_{l,V}', \text{HASH}_{l,W}'\}_{l=1}^L$ such that $\text{WL}_{\text{LP}}\left((G, H), L\right) \neq \text{WL}_{\text{LP}}\left((\hat{G}, \hat{H}), L\right)$. Unfortunately, there exist infinitely many pairs of non-isomorphic LP-graphs that cannot be distinguished by Algorithm 1. Figure 2 provide such an example.

Since the separation power of GNNs is actually equal to the WL test (Xu et al., 2019), one would expect that the limitation of the WL test might restrict GNNs from universally representing LP. However, any two LP-graphs that cannot be distinguished by the WL test must share some common characteristics even if they are not isomorphic. For example, let us consider the six LP instances in Figure 2. In each of the three columns, the two non-isomorphic LP instances cannot be distinguished by the WL test. It can be checked that the two instances in the same column share some common characteristics. More specifically, both instances in the first column are infeasible; both instances in the second column are feasible but unbounded; both instances in the third column are feasible and bounded with $(1/2, 1/2, 1/2, 1/2)$ being the optimal solution with the smallest $\ell_2$-norm. Actually, this phenomenon does not only happen on the instances in Figure 2, but also serves as an universal principle for all LP instances. We summarize the results in the following theorem:

**Theorem 4.1.** *If $(G, H), (\hat{G}, \hat{H}) \in \mathcal{G}_{m,n} \times \mathcal{H}_m^V \times \mathcal{H}_n^W$ are not distinguishable by Algorithm 1, then*

$$\Phi_{feas}(G, H) = \Phi_{feas}(\hat{G}, \hat{H}) \quad and \quad \Phi_{obj}(G, H) = \Phi_{obj}(\hat{G}, \hat{H}).$$

---

[2]In Algorithm 1, multisets, denoted by $\{\{\}\}$, are collections of elements that allow multiple appearance of the same element. Hash functions $\{\text{HASH}_{l,V}, \text{HASH}_{l,W}\}_{l=0}^L$ injectively map vertex information to vertex colors, while the others $\{\text{HASH}_{l,V}', \text{HASH}_{l,W}'\}_{l=1}^L$ injectively map vertex colors to a linear space so that one can define sum and scalar multiplications on their outputs. In addition, such an algorithm is usually named as the 1-WL test in the literature since it only considers the neighborhood with distance 1 for each vertex. In this paper, we abbreviate Algorithm 1 or the 1-WL test to the WL test for simplicity.

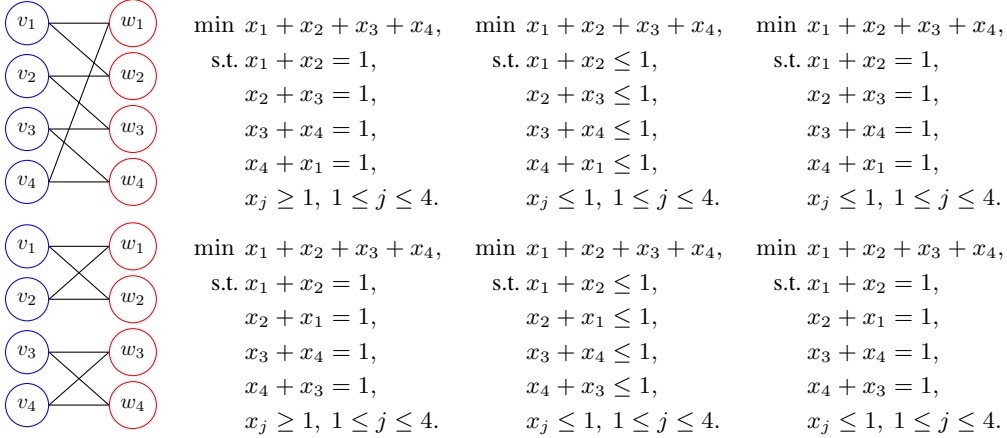

Figure 2: LP-graphs that cannot be distinguished by the WL test. Since the features and neighbor information of $\{v_i\}$ and $\{w_j\}$ in the two graphs are equal, it holds for both graphs that $C_1^{l,V} = \cdots C_4^{l,V}$ and $C_1^{l,W} = \cdots C_4^{l,W}$ for all $l \geq 0$, whatever the hash functions are chosen. Based on this graph pair, we construct three pairs of LPs that are both infeasible, both unbounded, both feasible bounded with the same optimal solution, respectively.

*Furthermore, if $(G, H), (\hat{G}, \hat{H}) \in \Phi_{obj}^{-1}(\mathbb{R})$, then it holds that $\Phi_{solu}(G, H) = \sigma_W(\Phi_{solu}(\hat{G}, \hat{H}))$ for some $\sigma_W \in S_n$.*

In other words, the above theorem guarantees the sufficient power of the WL test for separating LP problems with different characteristics, including feasibility, optimal objective value, and optimal solution with smallest $\ell_2$-norm (up to permutation). This combined with the following theorem, which states the equivalence of the separation powers of the WL test and GNNs, yields that GNNs also have sufficient separation power for LP-graphs in the above sense.

**Theorem 4.2.** *For any $(G, H), (\hat{G}, \hat{H}) \in \mathcal{G}_{m,n} \times \mathcal{H}_m^V \times \mathcal{H}_n^W$, the followings are equivalent:*

*(i) $(G, H)$ and $(\hat{G}, \hat{H})$ are not distinguishable by Algorithm 1.*

*(ii) $F(G, H) = F(\hat{G}, \hat{H}), \forall F \in \mathcal{F}_{GNN}$.*

*(iii) For any $F_W \in \mathcal{F}_{GNN}^W$, there exists $\sigma_W \in S_n$ such that $F_W(G, H) = \sigma_W(F_W(\hat{G}, \hat{H}))$.*

Theorem 4.2 extends the results in Xu et al. (2019); Azizian & Lelarge (2021); Geerts & Reutter (2022) to the case with the modified WL test (Algorithm 1) and LP-graphs.

**Representation power** Based on the separation power of GNNs, one is able to investigate the representation/approximation power of GNNs. To prove Theorems 3.2, 3.4, and 3.6, we first determine the closure of $\mathcal{F}_{GNN}$ or $\mathcal{F}_{GNN}^W$ in the space of invariant/equivariant continuous functions with respect to the sup-norm, which is also named as the *universal approximation*. The result of $\mathcal{F}_{GNN}$ is stated as follows, where $\mathcal{C}(X, \mathbb{R})$ is the collection/algebra of all real-valued continuous function on $X$. The result of $\mathcal{F}_{GNN}^W$ can be found in the appendix.

**Theorem 4.3.** *Let $X \subset \mathcal{G}_{m,n} \times \mathcal{H}_m^V \times \mathcal{H}_n^W$ be a compact set. For any $\Phi \in \mathcal{C}(X, \mathbb{R})$ that satisfies $\Phi(G, H) = \Phi(\hat{G}, \hat{H})$ for all $(G, H), (\hat{G}, \hat{H}) \in X$ that are not distinguishable by Algorithm 1, and any $\epsilon > 0$, there exists $F \in \mathcal{F}_{GNN}$ such that*

$$\sup_{(G,H) \in X} |\Phi(G, H) - F(G, H)| < \epsilon.$$

Theorem 4.3 can be viewed as an LP-graph version of results in Azizian & Lelarge (2021); Geerts & Reutter (2022). Roughly speaking, graph neural networks can approximate any invariant continuous function whose separation power is upper bounded by that of WL test on compact domain with arbitrarily small error. Although our target mappings $\Phi_{feas}, \Phi_{obj}, \Phi_{solu}$ are not continuous, we prove (in appendix) that they are measurable. Applying Lusin's theorem (Evans & Garzepy, 2018, Theorem 1.14), we show that GNN can be arbitrarily close to the target mappings except for a small domain.

## 5 NUMERICAL EXPERIMENTS

We present the numerical results that validate our theoretical results in this section. We generate LP instances with $m = 10$ and $n = 50$ that are possibly infeasible or feasible and bounded. To check whether GNN can predict feasibility, we generate three data sets with $100, 500, 2500$ independent LP instances respectively, and call the solver wrapped in `scipy.optimize.linprog` to get the feasibility, optimal objective value and an optimal solution for each generated LP. To generate enough feasible and bounded LPs to check whether GNN can approximate the optimal objective value and optimal solution, we follow the same approach as before to generate LP randomly and discard those infeasible LPs until the number of LPs reach our requirement. We train GNNs to fit the three LP characteristics by minimizing the distance between GNN-output and those solver-generated labels. The building and the training of the GNNs are implemented using `TensorFlow`. The codes are modified from Gasse et al. (2019) and can be found in `https://github.com/liujl11git/GNN-LP.git`. We set $L = 2$ for all GNNs and those learnable functions $f_{\text{in}}^V, f_{\text{in}}^W, f_{\text{out}}^V, f_{\text{out}}^W, \{f_l^V, f_l^W, g_l^V, g_l^W\}_{l=0}^L$ are all parameterized with MLPs. Details can be found in the appendix. Our results are reported in Figure 3.

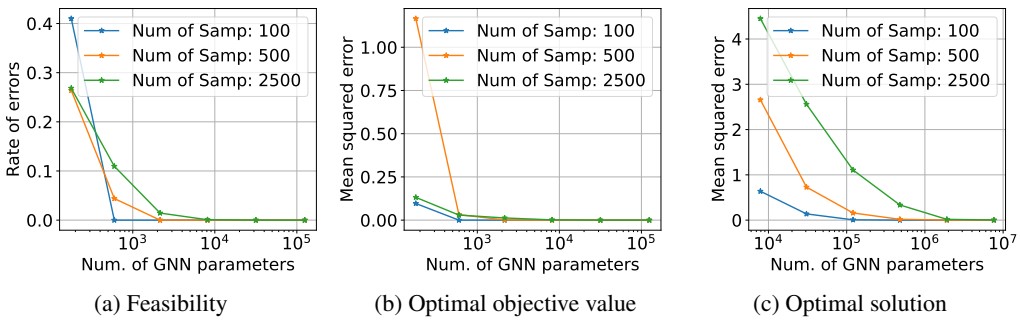

(a) Feasibility      (b) Optimal objective value      (c) Optimal solution

Figure 3: GNN can approximate $\Phi_{\text{feas}}$, $\Phi_{\text{obj}}$, and $\Phi_{\text{solu}}$

All the errors reported in Figure 3 are training errors since generalization is out of the scope of this paper. In Figure 3a, the "rate of errors" means the proportion of instances with $\mathbb{I}_{F(G,H)>1/2} \neq \Phi_{\text{feas}}(G, H)$. This metric exactly equals to zeros as long as the number of parameters in GNN is large enough, which directly validates Corollary 3.3: the existence of GNNs that can accurately predict the feasibility of LP instances. With the three curves in Figure 3a combined together, we conclude that such principle does not violate as the number of samples increases. This consists with Theorem 3.2. Mean squared errors in Figures 3b and 3c are respectively defined as $\mathbb{E}_{(G,H)}|F(G, H) - \Phi_{\text{feas}}(G, H)|^2$ and $\mathbb{E}_{(G,H)}\|F(G, H) - \Phi_{\text{feas}}(G, H)\|^2$. Therefore, Figures 3b and 3c validates Theorems 3.4 and 3.6 respectively. Note that all the instances used in Figure 3b are feasible and bounded. Thus, Figure 3b actually only validates (3.2) in Theorem 3.4. However, due to the fact that feasibility of an LP is equal to the boundedness of its dual problem, one may dualize each LP and use the conclusion of Figure 3a to validate (3.1) in Theorem 3.4. Some extra experimental results on generalization, i.e., the performance of the trained models on the test set, are presented in Appendix G.

## 6 CONCLUSIONS

In this work, we show that graph neural networks, as well as the WL test, have sufficient separation power to distinguish linear programming problems with different characteristics. In addition, GNNs can approximate LP feasibility, optimal objective value, and optimal solution with arbitrarily small errors on compact domains or finite datasets. These results guarantee that GNN is a proper class of machine learning models to represent linear programs, and hence contribute to the theoretical foundation in the learning-to-optimize community. Future directions include the size/complexity of GNNs and the generalization, that are not covered in our current theory but are of great importance. Another future topic is investing the representation power of graph neural networks for mixed-integer linear programming (MILP), which has been observed with promising experimental results in the literature.

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

## A   WEISFEILER-LEHMAN (WL) TEST AND COLOR REFINEMENT

The WL test can be viewed as a coloring refinement procedure if there are no collisions of hash functions and their weighted averages. More specifically, each vertex is colored initially according to the group it belongs to and its feature – two vertices have the same color if and only if they are in the same vertex group and have the same feature. The initial colors are denoted as $C_1^{0,V}, C_2^{0,V}, \ldots, C_m^{0,V}, C_1^{0,W}, C_2^{0,W}, \ldots, C_n^{0,W}$. Then at iteration $l$, the set of vertices with the same color at iteration $l-1$ are further partitioned into several subsets according to the colors of their neighbours – two vertices $v_i$ and $v_{i'}$ are in the same subset if and only if $C_i^{l-1,V} = C_{i'}^{l-1,V}$ and for any $C \in \{C_j^{l-1,W} : 1 \le j \le n\}$,

$$\sum_{C_j^{l-1,W}=C} E_{i,j} = \sum_{C_j^{l-1,W}=C} E_{i',j},$$

and it is similar for vertices $w_j$ and $w_{j'}$. After such partition/refinement, vertices are associated with the same color if and only if they are in the same subset, which is the coloring at iteration $l$. This procedure is terminated if the refinement is trivial, meaning that no sets with the same color are partitioned into at least two subsets, i.e., the coloring is stable. For more information about color refinement, we refer to Berkholz et al. (2017); Arvind et al. (2015; 2017).

We then discuss the stable coloring that Algorithm 1 will converge to, for which we made the following definition, where $\mathcal{S} = \{S_1, S_2, \ldots, S_s\}$ is called a partition of a set $S$ if $S_1 \cup S_2 \cup \cdots \cup S_s = S$ and $S_i \cap S_{i'} = \emptyset, \forall 1 \le i < i' \le s$.

**Definition A.1** (Stable Partition Pair of Vertices). *Let $G = (V \cup W, E)$ be a weighted bipartite graph with $V = \{v_1, v_2, \ldots, v_m\}$, $W = \{w_1, w_2, \ldots, w_n\}$, and vertex features $H = (h_1^V, h_2^V, \ldots, h_m^V, h_1^W, h_2^W, \ldots, h_n^W)$, and let $\mathcal{I} = \{I_1, I_2, \ldots, I_s\}$ and $\mathcal{J} = \{J_1, J_2, \ldots, J_t\}$ be partitions of $\{1, 2, \ldots, m\}$ and $\{1, 2, \ldots, n\}$, respectively. We say that $(\mathcal{I}, \mathcal{J})$ is a stable partition pair of vertices for the graph $G$ if the followings are satisfied:*

*(i) $h_i^V = h_{i'}^V$ holds if $i, i' \in I_p$ for some $p \in \{1, 2, \ldots, s\}$.*

*(ii) $h_j^W = h_{j'}^W$ holds if $j, j' \in J_q$ for some $q \in \{1, 2, \ldots, t\}$.*

*(iii) For any $p \in \{1, 2, \ldots, s\}$, $q \in \{1, 2, \ldots, t\}$, and $i, i' \in I_p$, $\sum_{j \in J_q} E_{i,j} = \sum_{j \in J_q} E_{i',j}$.*

*(iv) For any $p \in \{1, 2, \ldots, s\}$, $q \in \{1, 2, \ldots, t\}$, and $j, j' \in J_q$, $\sum_{i \in I_p} E_{i,j} = \sum_{i \in I_p} E_{i,j'}$.*

We denote $(\mathcal{I}^l, \mathcal{J}^l)$ as the partition pair corresponding the coloring at iteration $l$ of Algorithm 1. Suppose that there are no collisions. Then it is clear that $(\mathcal{I}^{l+1}, \mathcal{J}^{l+1})$ is finer than $(\mathcal{I}^l, \mathcal{J}^l)$, denoted as $(\mathcal{I}^{l+1}, \mathcal{J}^{l+1}) \preceq (\mathcal{I}^l, \mathcal{J}^l)$, which means that for any $I \in \mathcal{I}^{l+1}$ and any $J \in \mathcal{J}^{l+1}$, there exist $I' \in \mathcal{I}^l$ and $J' \in \mathcal{J}^l$ such that $I \subset I'$ and $J \subset J'$. In addition, $(\mathcal{I}^{l+1}, \mathcal{J}^{l+1}) = (\mathcal{I}^l, \mathcal{J}^l)$ if and only if $(\mathcal{I}^l, \mathcal{J}^l)$ is a stable partition pair of vertices. Note that there is at most $O(|V| + |W|)$ iterations leading to strictly finer partition pair, i.e, $(\mathcal{I}^{l+1}, \mathcal{J}^{l+1}) \preceq (\mathcal{I}^l, \mathcal{J}^l)$ but $(\mathcal{I}^{l+1}, \mathcal{J}^{l+1}) \neq (\mathcal{I}^l, \mathcal{J}^l)$. We can immediately obtain the following result:

**Theorem A.2.** *If there are no collision of hash functions and their weighted averages, then Algorithm 1 terminates at a stable partition pair of vertices in $O(|V| + |W|)$ iterations.*

Furthermore, for every $(G, H) \in \mathcal{G}_{m,n} \times \mathcal{H}_m^V \times \mathcal{H}_n^W$, the coarsest stable partition pair of vertices exists and is unique, which can be proved using techniques similar to the proof of Berkholz et al. (2017, Proposition 3). Algorithm 1 terminates at the unique coarsest stable partition pair. This is because that the coloring gin each iteration of Algorithm 1 is always coarser than the unique coarsest stable partition pair (see Berkholz et al. (2017, Proposition 2)).

## B   SEPARATION POWER OF THE WL TEST

This section gives the proof and some corollaries of Theorem 4.1. First we present some definitions and lemmas, from which the proof of Theorem 4.1 can be immediately derived.

**Definition B.1.** *Given* $(G, H), (\hat{G}, \hat{H}) \in \mathcal{G}_{m,n} \times \mathcal{H}_m^V \times \mathcal{H}_n^W$, *we say that* $(G, H)$ *and* $(\hat{G}, \hat{H})$ *can be distinguished by the WL test if there exists some* $L \in \mathbb{N}$ *and some choices of hash functions,* $HASH_{0,V}$, $HASH_{0,W}$, $HASH_{l,V}$, $HASH_{l,W}$, $HASH'_{l,V}$, *and* $HASH'_{l,W}$, *for* $l = 1, 2, \ldots, L$, *such that the multisets of colors at the L-th iteration of the WL test are different for* $(G, H)$ *and* $(\hat{G}, \hat{H})$. *Let* $\sim$ *be an equivalence relationship on* $\mathcal{G}_{m,n} \times \mathcal{H}_m^V \times \mathcal{H}_n^W$ *defined via:* $(G, H) \sim (\hat{G}, \hat{H})$ *if and only if they can not be distinguished by the WL test.*

It is clear that $(G, H) \sim (\hat{G}, \hat{H})$ if they are isometric, i.e., there exist two permutations, $\sigma_V : \{1, 2, \ldots, m\} \to \{1, 2, \ldots, m\}$ and $\sigma_W : \{1, 2, \ldots, n\} \to \{1, 2, \ldots, n\}$, such that $E_{\sigma_V(i), \sigma_W(j)} = \hat{E}_{i,j}$, $h_{\sigma_V(i)}^V = \hat{h}_i^V$, and $h_{\sigma_W(j)}^W = \hat{h}_j^W$, for any $i \in \{1, 2, \ldots, m\}$ and $j \in \{1, 2, \ldots, n\}$. However, not every pair of WL-indistinguishable graphs consists of isometric ones; see Figure 2 for an example. However, for LP problems that cannot be distinguished by the WL test will share some common properties, even if their associated graphs are not isomorphic.

**Lemma B.2.** *If two weighted bipartite graphs with vertex features corresponding to two LP problems are indistinguishable by the WL test, then either both problems are feasible or both are infeasible. In other words, the WL test can distinguish two LP problems if one of them is feasible while the other one is infeasible.*

*Proof of Lemma B.2.* Let us consider two LP problems:

$$\min_{x \in \mathbb{R}^n} \quad c^\top x,$$
$$\text{s.t.} \quad Ax \circ b, \, l \le x \le u, \tag{B.1}$$

and

$$\min_{x \in \mathbb{R}^n} \quad \hat{c}^\top x,$$
$$\text{s.t.} \quad \hat{A}x \, \hat{\circ} \, \hat{b}, \, \hat{l} \le x \le \hat{u}. \tag{B.2}$$

Let $(G, H)$ and $(\hat{G}, \hat{H})$, where $G = (V \cup W, E)$ and $\hat{G} = (V \cup W, \hat{E})$, be the weighted bipartite graphs with vertex features corresponding to (B.1) and (B.2), respectively.

Since $(G, H) \sim (\hat{G}, \hat{H})$, for some choice of hash functions with no collision during Algorithm 1 for $(G, H)$ and $(\hat{G}, \hat{H})$, Theorem A.2 guarantees that Algorithm 1 outputs the same stable coloring for $(G, H)$ and $(\hat{G}, \hat{H})$ up to permutation. More specifically, after doing some permutation, there exist $\mathcal{I} = \{I_1, I_2, \ldots, I_s\}$ and $\mathcal{J} = \{J_1, J_2, \ldots, J_t\}$ that are partitions of $\{1, 2, \ldots, m\}$ and $\{1, 2, \ldots, n\}$, respectively, such that the followings hold:

- $(b_i, \circ_i) = (\hat{b}_i, \hat{\circ}_i)$ and is independent of $i \in I_p$, for any $p \in \{1, 2, \ldots, s\}$.

- $(l_j, u_j) = (\hat{l}_j, \hat{u}_j)$ and is independent of $j \in J_q$ for any $q \in \{1, 2, \ldots, t\}$.

- For any $p \in \{1, 2, \ldots, s\}$ and $q \in \{1, 2, \ldots, t\}$, $\sum_{j \in J_q} A_{i,j} = \sum_{j \in J_q} \hat{A}_{i,j}$ and is independent of $i \in I_p$.

- For any $p \in \{1, 2, \ldots, s\}$ and $q \in \{1, 2, \ldots, t\}$, $\sum_{i \in I_p} A_{i,j} = \sum_{i \in I_p} \hat{A}_{i,j}$ and is independent of $j \in J_q$.

Suppose that the problem (B.1) is feasible with $x \in \mathbb{R}^n$ be some point in the feasible region. Define $y \in \mathbb{R}^t$ via $y_q = \frac{1}{|J_q|} \sum_{j \in J_q} x_j$ and $\hat{x} \in \mathbb{R}^n$ via $\hat{x}_j = y_q, j \in J_q$. Fix any $p \in \{1, 2, \ldots, s\}$ and some $i_0 \in I_p$. It holds for any $i \in I_p$ that

$$\sum_{j=1}^n A_{i,j} x_j \circ_i b_i, \quad \text{i.e.,} \quad \sum_{q=1}^t \sum_{j \in J_q} A_{i,j} x_j \circ_{i_0} b_{i_0},$$

which implies that

$$\frac{1}{|I_p|} \sum_{i \in I_p} \sum_{q=1}^t \sum_{j \in J_q} A_{i,j} x_j = \frac{1}{|I_p|} \sum_{q=1}^t \sum_{j \in J_q} \left( \sum_{i \in I_p} A_{i,j} \right) x_j \circ_{i_0} b_{i_0}.$$

Notice that $\sum_{i \in I_p} A_{i,j}$ is constant for $j \in J_q, \forall q \in \{1, 2, \ldots, t\}$. Let us denote $\alpha_q = \sum_{i \in I_p} A_{i,j} = \sum_{i \in I_p} \hat{A}_{i,j}$ for any $j \in J_q$. Then it holds that

$$\frac{1}{|I_p|} \sum_{q=1}^{t} \sum_{j \in J_q} \alpha_q x_j = \frac{1}{|I_p|} \sum_{q=1}^{t} \sum_{j \in J_q} \alpha_q y_q = \frac{1}{|I_p|} \sum_{q=1}^{t} \sum_{j \in J_q} \left( \sum_{i \in I_p} \hat{A}_{i,j} \right) y_q \circ_{i_0} b_{i_0}.$$

Note that

$$\frac{1}{|I_p|} \sum_{q=1}^{t} \sum_{j \in J_q} \left( \sum_{i \in I_p} \hat{A}_{i,j} \right) y_q = \frac{1}{|I_p|} \sum_{i \in I_p} \sum_{q=1}^{t} \left( \sum_{j \in J_q} \hat{A}_{i,j} \right) y_q,$$

and that $\sum_{j \in J_q} \hat{A}_{i,j}$ is constant for $i \in I_p$. So one can conclude that

$$\sum_{j=1}^{n} \hat{A}_{i,j} \hat{x}_j = \sum_{q=1}^{t} \sum_{j \in J_q} \hat{A}_{i,j} \hat{x}_j = \sum_{q=1}^{t} \left( \sum_{j \in J_q} \hat{A}_{i,j} \right) y_q \circ_i b_i, \quad \forall i \in I_p,$$

which leads to $\hat{A}\hat{x} \circ \hat{b}$. It can also be seen that $\hat{l} = l \le \hat{x} \le u \le \hat{u}$. Therefore, $\hat{x}$ is feasible for (B.2).

We have shown above that the feasibility of (B.1) implies the feasibility of (B.2). The inverse is also true by the same reasoning. Hence, we complete the proof. $\square$

**Lemma B.3.** *If two weighted bipartite graphs with vertex features corresponding to two LP problems are indistinguishable by the WL test, then these two problems share the same optimal objective value (could be $\infty$ or $-\infty$).*

*Proof of Lemma B.3.* If both problems are infeasible, then their optimal objective values are both $\infty$. We then consider the case that both problems are feasible. We use the same setting and notations as in Lemma B.2, and in addition we have that $c_j = \hat{c}_j$, which is part of $h_j^W = \hat{h}_j^W$, is independent of $j \in J_q$ for any $q \in \{1, 2, \ldots, t\}$. Suppose that $x$ is an feasible solution to the problem (B.1) and let $\hat{x} \in \mathbb{R}^n$ be defined via $\hat{x}_j = \frac{1}{|J_q|} \sum_{j' \in J_q} x_{j'}$, $j \in J_q$. It is guaranteed by the proof of Lemma B.2 that $\hat{x}$ is a feasible solution to (B.2). One can also see that $c^\top x = \hat{c}^\top \hat{x}$. Since this holds for any feasible solution $x$ to (B.1), the optimal value of the objective function for (B.2) is smaller than or equal to that for (B.1). The inverse is also true and the proof is completed. $\square$

**Lemma B.4.** *Suppose that two weighted bipartite graphs with vertex features corresponding to two LP problems are indistinguishable by the WL test and the their optimal objective values are both finite. Then these two problems have the same optimal solution with the smallest $\ell_2$-norm, up to permutation.*

*Proof of Lemma B.4.* We work with the same setting as in Lemma B.3, where permutations have already been applied. Let $x$ and $x'$ be the optimal solution to (B.1) and (B.2) with the smallest $\ell_2$-norm, respectively. (Recall that the optimal solution to a LP problem with the smallest $\ell_2$-norm is unique, see Remark 2.2.) Let $\hat{x} \in \mathbb{R}^n$ be defined via $\hat{x}_j = \frac{1}{|J_q|} \sum_{j' \in J_q} x_{j'}$ for $j \in J_q$, $q = 1, 2, \ldots, t$. According to the arguments in the proof of Lemma B.2 and Lemma B.3, $\hat{x}$ is an optimal solution to (B.2). The minimality of $x'$ yields that

$$\|x'\|_2^2 \le \|\hat{x}\|_2^2 = \sum_{q=1}^{t} |J_q| \left( \frac{1}{|J_q|} \sum_{j \in J_q} x_j \right)^2 = \sum_{q=1}^{t} \frac{1}{|J_q|} \left( \sum_{j \in J_q} x_j \right)^2 \le \sum_{q=1}^{t} \sum_{j \in J_q} x_j^2 = \|x\|_2^2,$$
(B.3)

which implies $\|x'\| \le \|x\|$. The converse $\|x\| \le \|x'\|$ is also true. Therefore, we must have $\|x\| = \|x'\|$ and hence, the inequalities in (B.3) must hold as equalities. Then one can conclude that $x_j = x_{j'}$ for any $j, j' \in J_q$ and $q = 1, 2, \ldots, t$, which leads to $x = \hat{x}$. Furthermore, it follows from $\|x'\| = \|\hat{x}\|$ and the uniqueness of $x'$ (see Remark 2.2) that $x' = \hat{x} = x$, which completes the proof. $\square$

One corollary one can see from the proof of Lemma B.4 is that the components of the optimal solution with the smallest $\ell_2$-norm must be the same if the two corresponding vertices have the same color in the WL test.

**Corollary B.5.** *Let $(G, H)$ be a weighted bipartite graph with vertex features and let $x$ be the optimal solution to the corresponding LP problem with the smallest $\ell_2$-norm. Suppose that for some $j, j' \in \{1, 2, \ldots, n\}$, one has $C_j^{l,W} = C_{j'}^{l,W}$ for any $l \in \mathbb{N}$ and any choices of hash functions, then $x_j = x_{j'}$.*

Let us also define another equivalence relationship on $\mathcal{G}_{m,n} \times \mathcal{H}_m^V \times \mathcal{H}_n^W$ where colors of $w_1, w_2, \ldots, w_j$ with ordering (not just multisets) are considered:

**Definition B.6.** *Given $(G, H), (\hat{G}, \hat{H}) \in \mathcal{G}_{m,n} \times \mathcal{H}_m^V \times \mathcal{H}_n^W$, $(G, H)$ and $(\hat{G}, \hat{H})$ are not in the same equivalence class of $\overset{W}{\sim}$ if and only if there exist some $L \in \mathbb{N}$ and some hash functions $HASH_{0,V}$, $HASH_{0,W}$, $HASH_{l,V}$, $HASH_{l,W}$, $HASH'_{l,V}$, and $HASH'_{l,W}$, $l = 1, 2, \ldots, L$, such that $\{\{C_1^{L,V}, C_2^{L,V}, \ldots, C_m^{L,V}\}\} \neq \{\{\hat{C}_1^{L,V}, \hat{C}_2^{L,V}, \ldots, \hat{C}_m^{L,V}\}\}$ or $C_j^{l,W} \neq \hat{C}_j^{l,W}$ for some $j \in \{1, 2, \ldots, n\}$.*

It is clear that $(G, H) \overset{W}{\sim} (\hat{G}, \hat{H})$ implies $(G, H) \sim (\hat{G}, \hat{H})$. One can actually obtain a stronger version of Lemma B.4 given $(G, H) \overset{W}{\sim} (\hat{G}, \hat{H})$.

**Corollary B.7.** *Suppose that two weighted bipartite graphs with vertex features corresponding to two LP problems, $(G, H)$ and $(\hat{G}, \hat{H})$, satisfies that $(G, H) \overset{W}{\sim} (\hat{G}, \hat{H})$. Then these two problems have the same optimal solution with the smallest $\ell_2$-norm.*

*Proof of Corollary B.7.* The proof of Lemma B.4 still applies with the difference that there is no permutation on $\{w_1, w_2, \ldots, w_n\}$. □

*Proof of Theorem 4.1.* The proof of Theorem 4.1 follows immediately from Lemmas B.2, B.3 and B.4. □

## C SEPARATION POWER OF GRAPH NEURAL NETWORKS

This section aims to prove Theorem 4.2, i.e., the separation power of GNNs is equivalent to that of the WL test. Similar results can be found in previous literature, see e.g. Xu et al. (2019); Azizian & Lelarge (2021); Geerts & Reutter (2022). We first introduce some lemmas that can directly imply Theorem 4.2. The lemma below, similar to Xu et al. (2019, Lemma 2), states that the separation power of GNNs is at most that of the WL test.

**Lemma C.1.** *Let $(G, H), (\hat{G}, \hat{H}) \in \mathcal{G}_{m,n} \times \mathcal{H}_m^V \times \mathcal{H}_n^W$. If $(G, H) \sim (\hat{G}, \hat{H})$, then for any $F_W \in \mathcal{F}_{GNN}^W$, there exists a permutation $\sigma_W \in S_n$ such that $F_W(G, H) = \sigma_W(F_W(\hat{G}, \hat{H}))$.*

*Proof of Lemma C.1.* First we describe the sketch of our proof. The assumption $(G, H) \sim (\hat{G}, \hat{H})$ implies that, if we apply the WL test on $(G, H)$ and $(\hat{G}, \hat{H})$, the test results should be exactly the same whatever the hash functions in the WL test we choose. In the first step, we define a set of hash functions that are injective on all possible inputs. Second, we show that, if we apply an arbitrarily chosen GNN: $F_W \in \mathcal{F}_{\text{GNN}}^W$ on $(G, H)$ and $(\hat{G}, \hat{H})$, the vertex features of the two graphs are exactly the same up to permutation, given the fact that the WL test results are the same. Finally, it concludes that $F_W(G, H)$ should be the same with $F_W(\hat{G}, \hat{H})$ up to permutation.

Let us first define hash functions. We choose $HASH_{0,V}$ and $HASH_{0,W}$ that are injective on the following sets (not multisets) respectively:

$$\{h_1^V, \ldots, h_m^V, \hat{h}_1^V, \ldots, \hat{h}_m^V\} \text{ and } \{h_1^W, \ldots, h_n^W, \hat{h}_1^W, \ldots, \hat{h}_n^W\}.$$

Let $\{C_i^{l-1,V}\}_{i=1}^m$, $\{C_j^{l-1,W}\}_{j=1}^n$ and $\{\hat{C}_i^{l-1,V}\}_{i=1}^m$, $\{\hat{C}_j^{l-1,W}\}_{j=1}^n$ be the vertex colors in the $(l-1)$-th iteration $(1 \leq l \leq L)$ in the WL test for $(G, H)$ and $(\hat{G}, \hat{H})$ respectively. Define two sets (not multisets) that collect different colors:

$$\mathbf{C}_{l-1}^V = \{C_1^{l-1,V}, \ldots, C_m^{l-1,V}, \hat{C}_1^{l-1,V}, \ldots, \hat{C}_m^{l-1,V}\},$$

and
$$\mathbf{C}_{l-1}^{W} = \{C_1^{l-1,W}, \ldots, C_n^{l-1,W}, \hat{C}_1^{l-1,W}, \ldots, \hat{C}_n^{l-1,W}\}.$$

The hash function $\text{HASH}_{l,V}'$ and $\text{HASH}_{l,W}'$ are chosen such that the outputs are located in some linear spaces and that $\{\text{HASH}_{l,V}'(C) : C \in \mathbf{C}_{l-1}^{V}\}$ and $\{\text{HASH}_{l,W}'(C) : C \in \mathbf{C}_{l-1}^{W}\}$ are both linearly independent. Finally, we choose hash functions $\text{HASH}_{l,V}$ and $\text{HASH}_{l,W}$ such that $\text{HASH}_{l,V}$ is injective on the set (not multiset)

$$\left\{ \left( C_i^{l-1,V}, \sum_{j=1}^{n} E_{i,j} \text{HASH}_{l,W}' \left( C_j^{l-1,W} \right) \right) : 1 \leq i \leq m \right\}$$

$$\cup \left\{ \left( \hat{C}_i^{l-1,V}, \sum_{j=1}^{n} E_{i,j} \text{HASH}_{l,W}' \left( \hat{C}_j^{l-1,W} \right) \right) : 1 \leq i \leq m \right\},$$

and that $\text{HASH}_{l,W}$ is injective on the set (not multiset)

$$\left\{ \left( C_j^{l-1,W}, \sum_{i=1}^{n} E_{i,j} \text{HASH}_{l,V}' \left( C_i^{l-1,V} \right) \right) : 1 \leq j \leq n \right\}$$

$$\cup \left\{ \left( \hat{C}_j^{l-1,W}, \sum_{i=1}^{n} E_{i,j} \text{HASH}_{l,V}' \left( \hat{C}_i^{l-1,V} \right) \right) : 1 \leq j \leq n \right\}.$$

Those hash functions give the vertex colors at the next iteration ($l$-th layer): $\{C_i^{l,V}\}_{i=1}^{m}$, $\{C_j^{l,W}\}_{j=1}^{n}$ and $\{\hat{C}_i^{l,V}\}_{i=1}^{m}$, $\{\hat{C}_j^{l,W}\}_{j=1}^{n}$.

Consider any $F_W \in \mathcal{F}_{\text{GNN}}^{W}$ and let $\{h_i^{l-1,V}\}_{i=1}^{m}$, $\{h_j^{l-1,W}\}_{j=1}^{n}$ and $\{\hat{h}_i^{l-1,V}\}_{i=1}^{m}$, $\{\hat{h}_j^{l-1,W}\}_{j=1}^{n}$ be the vertex features in the $l$-th layer ($0 \leq l \leq L$) of the graph neural network $F_W$. (Update rule refers to equations (2.1),(2.2),(2.3),(2.5)) We aim to prove by induction that for any $l \in \{0, 1, \ldots, L\}$, the followings hold:

(i) $C_i^{l,V} = C_{i'}^{l,V}$ implies $h_i^{l,V} = h_{i'}^{l,V}$, for $1 \leq i, i' \leq m$;

(ii) $\hat{C}_i^{l,V} = \hat{C}_{i'}^{l,V}$ implies $\hat{h}_i^{l,V} = \hat{h}_{i'}^{l,V}$, for $1 \leq i, i' \leq m$;

(iii) $C_i^{l,V} = \hat{C}_{i'}^{l,V}$ implies $h_i^{l,V} = \hat{h}_{i'}^{l,V}$, for $1 \leq i, i' \leq m$;

(iv) $C_j^{l,W} = C_{j'}^{l,W}$ implies $h_j^{l,W} = h_{j'}^{l,W}$, for $1 \leq j, j' \leq n$;

(v) $\hat{C}_j^{l,W} = \hat{C}_{j'}^{l,W}$ implies $\hat{h}_j^{l,W} = \hat{h}_{j'}^{l,W}$, for $1 \leq j, j' \leq n$;

(vi) $C_j^{l,W} = \hat{C}_{j'}^{l,W}$ implies $h_j^{l,W} = \hat{h}_{j'}^{l,W}$, for $1 \leq j, j' \leq n$.

The above claims (i)-(vi) are clearly true for $l = 0$ due to the injectivity of $\text{HASH}_{0,V}$ and $\text{HASH}_{0,W}$. Now we assume that (i)-(vi) are true for some $l - 1 \in \{0, 1, \ldots, L-1\}$. Suppose that $C_i^{l,V} = C_{i'}^{l,V}$, i.e.,

$$\text{HASH}_{l,V} \left( C_i^{l-1,V}, \sum_{j=1}^{n} E_{i,j} \text{HASH}_{l,W}' \left( C_j^{l-1,W} \right) \right)$$

$$= \text{HASH}_{l,V} \left( C_{i'}^{l-1,V}, \sum_{j=1}^{n} E_{i',j} \text{HASH}_{l,W}' \left( C_j^{l-1,W} \right) \right),$$

for some $1 \leq i, i' \leq m$. It follows from the injectivity of $\text{HASH}_{l,V}$ that

$$C_i^{l-1,V} = C_{i'}^{l-1,V}, \tag{C.1}$$

and

$$\sum_{j=1}^{n} E_{i,j} \text{HASH}'_{l,W} \left( C_j^{l-1,W} \right) = \sum_{j=1}^{n} E_{i',j} \text{HASH}'_{l,W} \left( C_j^{l-1,W} \right).$$

According to the linearly independent property of $\text{HASH}'_{l,W}$, the above equation implies that

$$\sum_{C_j^{l-1,W}=C} E_{i,j} = \sum_{C_j^{l-1,W}=C} E_{i',j}, \quad \forall\, C \in \mathbf{C}_{l-1}^{W}. \tag{C.2}$$

Note that the induction assumption guarantees that $h_j^{l-1,W} = h_{j'}^{l-1,W}$ as long as $C_j^{l-1,W} = C_{j'}^{l-1,W}$. So one can assign for each $C \in \mathbf{C}_{l-1}^{W}$ some $h(C) \in \mathbb{R}^{d_{l-1}}$ such that $h_j^{l-1,W} = h(C)$ as long as $C_j^{l-1,W} = C$ for any $1 \le j \le n$. Therefore, it follows from (C.2) that

$$\sum_{j=1}^{n} E_{i,j} f_l^{W}(h_j^{l-1,W}) = \sum_{C \in \mathbf{C}_{l-1}^{W}} \sum_{C_j^{l-1,W}=C} E_{i,j} f_l^{W}(h(C))$$

$$= \sum_{C \in \mathbf{C}_{l-1}^{W}} \sum_{C_j^{l-1,W}=C} E_{i',j} f_l^{W}(h(C)) = \sum_{j=1}^{n} E_{i',j} f_l^{W}(h_j^{l-1,W}).$$

Note also that (C.1) and the induction assumption lead to $h_i^{l-1,V} = h_{i'}^{l-1,V}$. Then one can conclude that

$$h_i^{l,V} = g_l^{V} \left( h_i^{l-1,V}, \sum_{j=1}^{n} E_{i,j} f_l^{W}(h_j^{l-1,W}) \right) = g_l^{V} \left( h_{i'}^{l-1,V}, \sum_{j=1}^{n} E_{i',j} f_l^{W}(h_j^{l-1,W}) \right) = h_{i'}^{l,V}.$$

This proves the claim (i) for $l$. The other five claims can be proved using similar arguments.

Therefore, we obtain from $(G, H) \sim (\hat{G}, \hat{H})$ that

$$\left\{ \left\{ h_1^{L,V}, h_2^{L,V}, \ldots, h_m^{L,V} \right\} \right\} = \left\{ \left\{ \hat{h}_1^{L,V}, \hat{h}_2^{L,V}, \ldots, \hat{h}_m^{L,V} \right\} \right\},$$

and that

$$\left\{ \left\{ h_1^{L,W}, h_2^{L,W}, \ldots, h_n^{L,W} \right\} \right\} = \left\{ \left\{ \hat{h}_1^{L,W}, \hat{h}_2^{L,W}, \ldots, \hat{h}_n^{L,W} \right\} \right\}.$$

By the definition of the output layer, the above conclusion guarantees that $F_W(G, H) = \sigma_W(F_W(\hat{G}, \hat{H}))$ for some $\sigma_W \in S_n$. $\qquad\square$

**Lemma C.2.** *Let $(G, H), (\hat{G}, \hat{H}) \in \mathcal{G}_{m,n} \times \mathcal{H}_m^{V} \times \mathcal{H}_n^{W}$. Suppose that for any $F_W \in \mathcal{F}_{GNN}^{W}$, there exists a permutation $\sigma_W \in S_n$ such that $F_W(G, H) = \sigma_W(F_W(\hat{G}, \hat{H}))$. Then $F(G, H) = F(\hat{G}, \hat{H})$ holds for any $F \in \mathcal{F}_{GNN}$.*

*Proof of Lemma C.2.* Pick an arbitrary $F \in \mathcal{F}_{\text{GNN}}$. We choose $F_W \in \mathcal{F}_{\text{GNN}}^{W}$ such that

$$F_W(G', H') = (F(G', H'), \ldots, F(G', H'))^{\top} \in \mathbb{R}^{n}, \quad \forall\, (G', H') \in \mathcal{G}_{m,n} \times \mathcal{H}_m^{V} \times \mathcal{H}_n^{W}.$$

Note that every entry in the output of $F_W$ is equal to the output of $F$. Thus, it follows from $F_W(G, H) = \sigma_W(F_W(\hat{G}, \hat{H}))$ that $F(G, H) = F(\hat{G}, \hat{H})$. $\qquad\square$

The next lemma is similar to Xu et al. (2019, Theorem 3) and states that the separation power of GNNs is at least that of the WL test.

**Lemma C.3.** *Let $(G, H), (\hat{G}, \hat{H}) \in \mathcal{G}_{m,n} \times \mathcal{H}_m^{V} \times \mathcal{H}_n^{W}$. If $F(G, H) = F(\hat{G}, \hat{H})$ holds for any $F \in \mathcal{F}_{GNN}$, then $(G, H) \sim (\hat{G}, \hat{H})$.*

*Proof of Lemma C.3.* It suffices to prove that, if $(G, H)$ can be distinguished from $(\hat{G}, \hat{H})$ by the WL test, then there exists $F \in \mathcal{F}_{\text{GNN}}$, such that $F(G, H) \neq F(\hat{G}, \hat{H})$. The distinguish-ability of the WL test implies that there exists $L \in \mathbb{N}$ and hash functions, $\text{HASH}_{0,V}$, $\text{HASH}_{0,W}$, $\text{HASH}_{l,V}$, $\text{HASH}_{l,W}$, $\text{HASH}'_{l,V}$, and $\text{HASH}'_{l,W}$, for $l = 1, 2, \ldots, L$, such that

$$\left\{\left\{C_1^{L,V}, C_2^{L,V}, \ldots, C_m^{L,V}\right\}\right\} \neq \left\{\left\{\hat{C}_1^{L,V}, \hat{C}_2^{L,V}, \ldots, \hat{C}_m^{L,V}\right\}\right\}, \tag{C.3}$$

or

$$\left\{\left\{C_1^{L,W}, C_2^{L,W}, \ldots, C_n^{L,W}\right\}\right\} \neq \left\{\left\{\hat{C}_1^{L,W}, \hat{C}_2^{L,W}, \ldots, \hat{C}_n^{L,W}\right\}\right\}, \tag{C.4}$$

We aim to construct some GNNs such that the followings hold for any $l = 0, 1, \ldots, L$:

(i) $h_i^{l,V} = h_{i'}^{l,V}$ implies $C_i^{l,V} = C_{i'}^{l,V}$, for $1 \leq i, i' \leq m$;

(ii) $\hat{h}_i^{l,V} = \hat{h}_{i'}^{l,V}$ implies $\hat{C}_i^{l,V} = \hat{C}_{i'}^{l,V}$, for $1 \leq i, i' \leq m$;

(iii) $h_i^{l,V} = \hat{h}_{i'}^{l,V}$ implies $C_i^{l,V} = \hat{C}_{i'}^{l,V}$, for $1 \leq i, i' \leq m$;

(iv) $h_j^{l,W} = h_{j'}^{l,W}$ implies $C_j^{l,W} = C_{j'}^{l,W}$, for $1 \leq j, j' \leq n$;

(v) $\hat{h}_j^{l,W} = \hat{h}_{j'}^{l,W}$ implies $\hat{C}_j^{l,W} = \hat{C}_{j'}^{l,W}$, for $1 \leq j, j' \leq n$;

(vi) $h_j^{l,W} = \hat{h}_{j'}^{l,W}$ implies $C_j^{l,W} = \hat{C}_{j'}^{l,W}$, for $1 \leq j, j' \leq n$.

It is clear that the above conditions (i)-(vi) hold for $l = 0$ as long as we choose $f_{\text{in}}^V$ and $f_{\text{in}}^W$ that are injective on the following two sets (not multisets) respectively:

$$\{h_1^V, \ldots, h_m^V, \hat{h}_1^V, \ldots, \hat{h}_m^V\} \text{ and } \{h_1^W, \ldots, h_n^W, \hat{h}_1^W, \ldots, \hat{h}_n^W\}.$$

We then assume that (i)-(vi) hold for some $0 \leq l - 1 < L$, and show that these conditions are also satisfied for $l$ if we choose $f_l^V, f_l^W, g_l^V, g_l^W$ properly. Let us consider the set (not multiset):

$$\{\alpha_1, \alpha_2, \ldots, \alpha_s\} \subset \mathbb{R}^{d_{l-1}}$$

that collects all different values in $h_1^{l-1,W}, h_2^{l-1,W}, \ldots, h_n^{l-1,W}, \hat{h}_1^{l-1,W}, \hat{h}_2^{l-1,W}, \ldots, \hat{h}_n^{l-1,W}$. Let $d_l \geq s$ and let $e_p^{d_l} = (0, \ldots, 0, 1, 0, \ldots, 0)$ be the vector in $\mathbb{R}^{d_l}$ with the $p$-th entry being 1 and all other entries being 0, for $1 \leq p \leq s$. Choose $f_l^W : \mathbb{R}^{d_{l-1}} \to \mathbb{R}^{d_l}$ as a continuous function satisfying $f_l^W(\alpha_p) = e_p^{d_l}$, $p = 1, 2, \ldots, s$, and choose $g_l^V : \mathbb{R}^{d_{l-1}} \times \mathbb{R}^{d_l} \to \mathbb{R}^{d_l}$ that is continuous and is injective when restricted on the set (not multiset)

$$\left\{\left(h_i^{l-1,V}, \sum_{j=1}^n E_{i,j} f_l^W(h_j^{l-1,W})\right) : 1 \leq i \leq m\right\}$$

$$\cup \left\{\left(\hat{h}_i^{l-1,V}, \sum_{j=1}^n \hat{E}_{i,j} f_l^W(\hat{h}_j^{l-1,W})\right) : 1 \leq i \leq m\right\}.$$

Noticing that

$$\sum_{j=1}^n E_{i,j} f_l^W(h_j^{l-1,W}) = \sum_{p=1}^s \left(\sum_{h_j^{l-1,W} = \alpha_p} E_{i,j}\right) e_p^{d_l},$$

and that $\{e_1^{d_l}, e_2^{d_l}, \ldots, e_s^{d_l}\}$ is linearly independent, one can conclude that $h_i^{l,V} = h_{i'}^{l,V}$ if and only if $h_i^{l-1,V} = h_{i'}^{l-1,V}$ and $\sum_{j=1}^n E_{i,j} f_l^W(h_j^{l-1,W}) = \sum_{j=1}^n E_{i',j} f_l^W(h_j^{l-1,W})$, where the second condition is equivalent to

$$\sum_{h_j^{l-1,W} = \alpha_p} E_{i,j} = \sum_{h_j^{l-1,W} = \alpha_p} E_{i',j}, \quad \forall p \in \{1, 2, \ldots, s\}.$$

This, as well as the condition (iv) for $l - 1$, implies that

$$\sum_{j=1}^{n} E_{i,j} \text{HASH}'_{l,W} \left( C_j^{l-1,W} \right) = \sum_{j=1}^{n} E_{i',j} \text{HASH}'_{l,W} \left( C_j^{l-1,W} \right),$$

and hence that $C_i^{l,V} = C_{i'}^{l,V}$ by using $h_i^{l-1,V} = h_i^{l-1,V}$ and condition (i) for $l - 1$. Therefore, we know that (i) is satisfied for $l$, and one can show (ii) and (iii) for $l$ using similar arguments by taking $d_l$ large enough. In addition, $f_l^V$ and $g_l^W$ can also be chosen in a similar way such that (iv)-(vi) are satisfied for $l$.

Combining (C.3), (C.4), and condition (i)-(iv) for $L$, we obtain that

$$\left\{ \left\{ h_1^{L,V}, h_2^{L,V}, \ldots, h_m^{L,V} \right\} \right\} \neq \left\{ \left\{ \hat{h}_1^{L,V}, \hat{h}_2^{L,V}, \ldots, \hat{h}_m^{L,V} \right\} \right\}, \tag{C.5}$$

or

$$\left\{ \left\{ h_1^{L,W}, h_2^{L,W}, \ldots, h_n^{L,W} \right\} \right\} \neq \left\{ \left\{ \hat{h}_1^{L,W}, \hat{h}_2^{L,W}, \ldots, \hat{h}_n^{L,W} \right\} \right\}.$$

Without loss of generality, we can assume that (C.5) holds.

Consider the set (not multiset)

$$\{\beta_1, \beta_2, \ldots, \beta_t\} \subset \mathbb{R}^{d_L},$$

that collects all different values in $h_1^{L,V}, h_2^{L,V}, \ldots, h_m^{L,V}, \hat{h}_1^{L,V}, \hat{h}_2^{L,V}, \ldots, \hat{h}_m^{L,V}$. Let $k > 1$ be a positive integer that is greater than the maximal multiplicity of an element in the multisets $\{\{h_1^{L,V}, h_2^{L,V}, \ldots, h_m^{L,V}\}\}$ and $\{\{\hat{h}_1^{L,V}, \hat{h}_2^{L,V}, \ldots, \hat{h}_m^{L,V}\}\}$. There exists a continuous function $\varphi : \mathbb{R}^{d_L} \to \mathbb{R}$ such that $\varphi(\beta_q) = k^q$ for $q = 1, 2, \ldots, t$, and due to (C.5) and the fact that the way of writing an integer as $k$-ary expression is unique, it hence holds that

$$\sum_{i=1}^{m} \varphi(h_i^{L,V}) \neq \sum_{i=1}^{m} \varphi(\hat{h}_i^{L,V}).$$

Set the dimension of $(L + 1)$-th layer as 1: $d_{L+1} = 1$, and set $f_{L+1,V} = 0$, $f_{L+1,W} = 0$, $g_{L+1,V}(h, 0) = \varphi(h)$, and $g_{L+1,W} = 0$. Then we have $h_i^{L+1,V} = \varphi(h_i^{L,V})$, $\hat{h}_i^{L+1,V} = \varphi(\hat{h}_i^{L,V})$, and $h_j^{L+1,W} = \hat{h}_j^{L+1,W}$ for $i = 1, 2, \ldots, m$ and $j = 1, 2, \cdots, n$. Define $f_{\text{out}} : \mathbb{R} \times \mathbb{R} \to \mathbb{R}$ via $f_{\text{out}}(h, h') = h$. Then it follows that

$$f_{\text{out}} \left( \sum_{i=1}^{m} h_i^{L+1,V}, \sum_{j=1}^{n} h_j^{L+1,W} \right) = \sum_{i=1}^{m} \varphi(h_i^{L,V})$$

$$\neq \sum_{i=1}^{m} \varphi(\hat{h}_i^{L,V}) = f_{\text{out}} \left( \sum_{i=1}^{m} \hat{h}_i^{L+1,V}, \sum_{j=1}^{n} \hat{h}_j^{L+1,W} \right),$$

which guarantees the existence of $F \in \mathcal{F}_{\text{GNN}}$ that has $L + 1$ layers and satisfies $F(G, H) \neq F(\hat{G}, \hat{H})$. $\qquad \square$

*Proof of Theorem 4.2.* The equivalence of the three conditions follow immediately from Lemma C.1, C.2, and C.3. $\qquad \square$

**Corollary C.4.** *For any two weighted bipartite graphs with vertex features* $(G, H), (\hat{G}, \hat{H}) \in \mathcal{G}_{m,n} \times \mathcal{H}_m^V \times \mathcal{H}_n^W$, *the followings are equivalent:*

*(i)* $(G, H) \overset{W}{\sim} (\hat{G}, \hat{H})$.

*(ii) For any* $F_W \in \mathcal{F}_{GNN}^W$, *it holds that* $F_W(G, H) = F_W(\hat{G}, \hat{H})$.

*Proof of Corollary C.4.* The proof follows similar lines as in the proof of Theorem 4.2 with the difference that there is no permutation on $\{w_1, w_2, \ldots, w_n\}$. $\qquad \square$

In addition to the separation power of GNNs for two weighted bipartite graphs with vertex features, one can also obtain results on separating different vertices in one weighted bipartite graph with vertex features.

**Corollary C.5.** *For any weighted bipartite graph with vertex features $(G, H)$ and any $j, j' \in \{1, 2, \ldots, n\}$, the followings are equivalent:*

*(i) $C_j^{l,W} = C_{j'}^{l,W}$ holds for any $l \in \mathbb{N}$ and any choice of hash functions.*

*(ii) $F_W(G, H)_j = F_W(G, H)_{j'}, \forall F_W \in \mathcal{F}_{GNN}^W$.*

*Proof of Corollary C.5.* "(i) $\implies$ (ii)" and "(ii) $\implies$ (i)" can be proved using similar arguments in the proof of Lemma C.1 and Lemma C.3, respectively. $\qquad\square$

# D  Universal Approximation of $\mathcal{F}_{\text{GNN}}$

This section provides the proof of Theorem 4.3. The main mathematical tool used in the proof is the Stone–Weierstrass theorem:

**Theorem D.1** (Stone–Weierstrass theorem (Rudin, 1991, Section 5.7))**.** *Let $X$ be a compact Hausdorff space and let $\mathcal{F} \subset \mathcal{C}(X, \mathbb{R})$ be a subalgebra. If $\mathcal{F}$ separates points on $X$, i.e., for any $x, x' \in X$ with $x \neq x'$, there exists $F \in \mathcal{F}$ such that $F(x) \neq F(x')$, and $1 \in \mathcal{F}$, then $\mathcal{F}$ is dense in $\mathcal{C}(X, \mathbb{R})$ with the topology of uniform convergence.*

*Proof of Theorem 4.3.* Let $\pi : X \to X/\sim$ be the quotient map, where $\pi(X) = X/\sim$ is equipped with the quotient topology. For any $F \in \mathcal{F}_{\text{GNN}}$, since $F : X \to \mathbb{R}$ is continuous and by Theorem 4.2, $F(G, H) = F(\hat{G}, \hat{H}), \forall (G, H) \sim (\hat{G}, \hat{H})$, there exists a unique continuous $\tilde{F} : \pi(X) \to \mathbb{R}$ such that $F = \tilde{F} \circ \pi$. Set
$$\tilde{\mathcal{F}}_{\text{GNN}} = \left\{ \tilde{F} : F \in \mathcal{F}_{\text{GNN}} \right\} \subset \mathcal{C}(\pi(X), \mathbb{R}).$$
In addition, the assumption on $\Phi$, i.e,
$$\Phi(G, H) = \Phi(\hat{G}, \hat{H}), \quad \forall (G, H) \sim (\hat{G}, \hat{H}),$$
leads to the existence of a unique $\tilde{\Phi} \in \mathcal{C}(\pi(X), \mathbb{R})$ with $\Phi = \tilde{\Phi} \circ \pi$.

Since $X$ is compact, then $\pi(X)$ is also compact due to the continuity of $\pi$. According to Lemma D.2 below, $\tilde{\mathcal{F}}_{\text{GNN}}$ is a subalgebra of $\mathcal{C}(\pi(X), \mathbb{R})$. By Theorem 4.2, $\tilde{\mathcal{F}}_{\text{GNN}}$ separates points on $\pi(X)$. This can further imply that $\pi(X)$ is Hausdorff. In fact, for any $x, x' \in \pi(X)$, there exists $\tilde{F} \in \tilde{F}_{\text{GNN}}$ with $\tilde{F}(x) \neq \tilde{F}(x')$. Without loss of generality, we assume that $\tilde{F}(x) < \tilde{F}(x')$ and choose some $c \in \mathbb{R}$ with $\tilde{F}(x) < c < \tilde{F}(x')$. By continuity of $\tilde{F}$, we know that $\tilde{F}^{-1}((-\infty, c)) \cap \pi(X)$ and $\tilde{F}^{-1}((c, +\infty)) \cap \pi(X)$ are disjoint open subsets of $\pi(X)$ with $x \in \tilde{F}^{-1}((-\infty, c)) \cap \pi(X)$ and $x' \in \tilde{F}^{-1}((c, +\infty)) \cap \pi(X)$, which leads to the Hausdorff property of $\pi(X)$. Note also that $1 \in \tilde{\mathcal{F}}_{\text{GNN}}$. Using Theorem D.1, we can conclude the denseness of $\tilde{\mathcal{F}}_{\text{GNN}}$ in $\mathcal{C}(\pi(X), \mathbb{R})$. Therefore, for any $\epsilon > 0$, there exists $F \in \mathcal{F}_{\text{GNN}}$, such that
$$\sup_{(G,H) \in X} |\Phi(G, H) - F(G, H)| = \sup_{x \in \pi(X)} |\tilde{\Phi}(x) - \tilde{F}(x)| < \epsilon,$$
which completes the proof. $\qquad\square$

**Lemma D.2.** *$\mathcal{F}_{GNN}$ is a subalgebra of $\mathcal{C}(\mathcal{G}_{m,n} \times \mathcal{H}_m^V \times \mathcal{H}_n^W, \mathbb{R})$, and as a corollary, $\tilde{\mathcal{F}}_{GNN}$ is a subalgebra of $\mathcal{C}(\mathcal{G}_{m,n} \times \mathcal{H}_m^V \times \mathcal{H}_n^W / \sim, \mathbb{R})$.*

*Proof of Lemma D.2.* It suffices to show that $\mathcal{F}_{\text{GNN}}$ is closed under addition and multiplication. Consider any $F, \hat{F} \in \mathcal{F}_{\text{GNN}}$. Thanks to Lemma D.3, we can assume that both $F$ and $\hat{F}$ have $L$ layers. Suppose that $F$ is constructed by
$$f_{\text{in}}^V : \mathcal{H}^V \to \mathbb{R}^{d_0}, \quad f_{\text{in}}^W : \mathcal{H}^W \to \mathbb{R}^{d_0},$$
$$f_l^V, f_l^W : \mathbb{R}^{d_{l-1}} \to \mathbb{R}^{d_l}, \quad g_l^V, g_l^W : \mathbb{R}^{d_{l-1}} \times \mathbb{R}^{d_l} \to \mathbb{R}^{d_l}, \quad 1 \leq l \leq L,$$
$$f_{\text{out}} : \mathbb{R}^{d_L} \times \mathbb{R}^{d_L} \to \mathbb{R},$$

and that $\hat{F}$ is constructed by

$$\hat{f}_{\text{in}}^V : \mathcal{H}^V \to \mathbb{R}^{\hat{d}_0}, \quad \hat{f}_{\text{in}}^W : \mathcal{H}^W \to \mathbb{R}^{\hat{d}_0},$$
$$\hat{f}_l^V, \hat{f}_l^W : \mathbb{R}^{\hat{d}_{l-1}} \to \mathbb{R}^{\hat{d}_l}, \quad \hat{g}_l^V, \hat{g}_l^W : \mathbb{R}^{\hat{d}_{l-1}} \times \mathbb{R}^{\hat{d}_l} \to \mathbb{R}^{\hat{d}_l}, \quad 1 \le l \le L,$$
$$\hat{f}_{\text{out}} : \mathbb{R}^{\hat{d}_L} \times \mathbb{R}^{\hat{d}_L} \to \mathbb{R}.$$

One can then construct two new GNNs computing $F + \hat{F}$ and $F \cdot \hat{F}$ as follows:

**The input layer** The update rule of the input layer is defined by

$$\mathbf{f}_{\text{in}}^V : \mathcal{H}^V \to \quad \mathbb{R}^{d_0} \times \mathbb{R}^{\hat{d}_0},$$
$$h \mapsto \left( f_{\text{in}}^V(h), \hat{f}_{\text{in}}^V(h) \right),$$
$$\mathbf{f}_{\text{in}}^W : \mathcal{H}^W \to \quad \mathbb{R}^{d_0} \times \mathbb{R}^{\hat{d}_0},$$
$$h \mapsto \left( f_{\text{in}}^W(h), \hat{f}_{\text{in}}^W(h) \right).$$

Then the vertex features after the computation of the input layer $\mathbf{h}_i^{0,V}$ and $\mathbf{h}_j^{0,W}$ are given by:

$$\mathbf{h}_i^{0,V} = \mathbf{f}_{\text{in}}^V(h_i^V) = \left( f_{\text{in}}^V(h_i^V), \hat{f}_{\text{in}}^V(h_i^V) \right) = (h_i^{0,V}, \hat{h}_i^{0,V}) \in \mathbb{R}^{d_0} \times \mathbb{R}^{\hat{d}_0},$$
$$\mathbf{h}_j^{0,W} = \mathbf{f}_{\text{in}}^W(h_j^W) = \left( f_{\text{in}}^W(h_j^W), \hat{f}_{\text{in}}^W(h_j^W) \right) = (h_j^{0,W}, \hat{h}_j^{0,W}) \in \mathbb{R}^{d_0} \times \mathbb{R}^{\hat{d}_0},$$

for $i = 1, 2 \dots, m$, and $j = 1, 2, \dots, n$.

**The $l$-th layer ($1 \le l \le L$)** . We set

$$\mathbf{f}_l^V : \mathbb{R}^{d_{l-1}} \times \mathbb{R}^{\hat{d}_{l-1}} \to \quad \mathbb{R}^{d_l} \times \mathbb{R}^{\hat{d}_l},$$
$$(h, \hat{h}) \mapsto \left( f_l^V(h), \hat{f}_l^V(\hat{h}) \right),$$
$$\mathbf{f}_l^W : \mathbb{R}^{d_{l-1}} \times \mathbb{R}^{\hat{d}_{l-1}} \to \quad \mathbb{R}^{d_l} \times \mathbb{R}^{\hat{d}_l},$$
$$(h, \hat{h}) \mapsto \left( f_l^W(h), \hat{f}_l^W(\hat{h}) \right),$$
$$\mathbf{g}_l^V : \mathbb{R}^{d_{l-1}} \times \mathbb{R}^{\hat{d}_{l-1}} \times \mathbb{R}^{d_l} \times \mathbb{R}^{\hat{d}_l} \to \quad \mathbb{R}^{d_l} \times \mathbb{R}^{\hat{d}_l},$$
$$(h, \hat{h}, h', \hat{h}') \mapsto \left( g_l^W(h, h'), \hat{g}_l^W(\hat{h}, \hat{h}') \right),$$

and

$$\mathbf{g}_l^W : \mathbb{R}^{d_{l-1}} \times \mathbb{R}^{\hat{d}_{l-1}} \times \mathbb{R}^{d_l} \times \mathbb{R}^{\hat{d}_l} \to \quad \mathbb{R}^{d_l} \times \mathbb{R}^{\hat{d}_l},$$
$$(h, \hat{h}, h', \hat{h}') \mapsto \left( g_l^V(h, h'), \hat{g}_l^V(\hat{h}, \hat{h}') \right).$$

Then the vertex features after the computation of the $l$-th layer $\mathbf{h}_i^{l,V}$ and $\mathbf{h}_j^{l,W}$ are given by:

$$\mathbf{h}_i^{l,V} = \mathbf{g}_l^V \left( \mathbf{h}_i^{l-1,V}, \sum_{j=1}^n E_{i,j} \mathbf{f}_l^W(\mathbf{h}_j^{l-1,W}) \right)$$
$$= \left( g_l^V \left( h_i^{l-1,V}, \sum_{j=1}^n E_{i,j} f_l^W(h_j^{l-1,W}) \right), \hat{g}_l^V \left( \hat{h}_i^{l-1,V}, \sum_{j=1}^n E_{i,j} \hat{f}_l^W(\hat{h}_j^{l-1,W}) \right) \right)$$
$$= (h_i^{l,V}, \hat{h}_i^{l,V}) \in \mathbb{R}^{d_l} \times \mathbb{R}^{\hat{d}_l},$$

and

$$\mathbf{h}_j^{l,W} = \mathbf{g}_l^W \left( \mathbf{h}_j^{l-1,W}, \sum_{i=1}^m E_{i,j} \mathbf{f}_l^V(\mathbf{h}_i^{l-1,V}) \right)$$

$$= \left( g_l^W \left( h_j^{l-1,W}, \sum_{i=1}^m E_{i,j} f_l^V(h_i^{l-1,V}) \right), \hat{g}_l^W \left( \hat{h}_j^{l-1,W}, \sum_{i=1}^m E_{i,j} \hat{f}_l^V(\hat{h}_i^{l-1,V}) \right) \right)$$

$$= (h_j^{l,W}, \hat{h}_j^{l,W}) \in \mathbb{R}^{d_l} \times \mathbb{R}^{\hat{d}_l},$$

for $i = 1, 2 \ldots, m$, and $j = 1, 2, \ldots, n$.

**The output layer**  To obtain $F + \hat{F}$, we set

$$\mathbf{f}_{\text{out}}^{\text{add}} : \mathbb{R}^{d_L} \times \mathbb{R}^{\hat{d}_L} \times \mathbb{R}^{d_L} \times \mathbb{R}^{\hat{d}_L} \rightarrow \mathbb{R},$$
$$((h, \hat{h}), (h', \hat{h}')) \mapsto f_{\text{out}}(h, h') + \hat{f}_{\text{out}}(\hat{h}, \hat{h}').$$

Then it holds that

$$\mathbf{f}_{\text{out}}^{\text{add}} \left( \sum_{i=1}^m \mathbf{h}_i^{L,V}, \sum_{j=1}^n \mathbf{h}_1^{L,W} \right) = f_{\text{out}} \left( \sum_{i=1}^m h_i^{L,V}, \sum_{j=1}^n h_j^{L,W} \right) + \hat{f}_{\text{out}} \left( \sum_{i=1}^m \hat{h}_i^{L,V}, \sum_{j=1}^n \hat{h}_j^{L,W} \right)$$
$$= F(G, H) + \hat{F}(G, H).$$

To obtain $F \cdot \hat{F}$, we set

$$\mathbf{f}_{\text{out}}^{\text{multiply}} : \mathbb{R}^{d_L} \times \mathbb{R}^{\hat{d}_L} \times \mathbb{R}^{d_L} \times \mathbb{R}^{\hat{d}_L} \rightarrow \mathbb{R},$$
$$((h, \hat{h}), (h', \hat{h}')) \mapsto f_{\text{out}}(h, h') \cdot \hat{f}_{\text{out}}(\hat{h}, \hat{h}').$$

Then it holds that

$$\mathbf{f}_{\text{out}}^{\text{multiply}} \left( \sum_{i=1}^m \mathbf{h}_i^{L,V}, \sum_{j=1}^n \mathbf{h}_1^{L,W} \right) = f_{\text{out}} \left( \sum_{i=1}^m h_i^{L,V}, \sum_{j=1}^n h_j^{L,W} \right) \cdot \hat{f}_{\text{out}} \left( \sum_{i=1}^m \hat{h}_i^{L,V}, \sum_{j=1}^n \hat{h}_j^{L,W} \right)$$
$$= F(G, H) \cdot \hat{F}(G, H).$$

The constructed GNNs satisfy $F(G, H) + \hat{F}(G, H) \in \mathcal{F}_{\text{GNN}}$ and $F(G, H) \cdot \hat{F}(G, H) \in \mathcal{F}_{\text{GNN}}$, which finishes the proof. □

**Lemma D.3.** *If $F \in \mathcal{F}_{GNN}$ has $L$ layers, then there exists $\hat{F} \in \mathcal{F}_{GNN}$ with $L + 1$ layers such that $F = \hat{F}$.*

*Proof of Lemma D.3.* Suppose that $F$ is constructed by $f_{\text{in}}^V, f_{\text{in}}^W, f_{\text{out}}, \{f_l^V, f_l^W, g_l^V, g_l^W\}_{l=0}^L$. We choose $f_{L+1}^V \equiv 0$, $f_{L+1}^W \equiv 0$, $g_{L+1}^V(h, h') = h$, $g_{L+1}^W(h, h') = h$. Let $\hat{F}$ be constructed by $f_{\text{in}}^V, f_{\text{in}}^W, f_{\text{out}}, \{f_l^V, f_l^W, g_l^V, g_l^W\}_{l=0}^{L+1}$. Then $\hat{F}$ has $L + 1$ layers with $\hat{F} = F$. □

# E  UNIVERSAL APPROXIMATION OF $\mathcal{F}_{\text{GNN}}^W$

This section provides an universal approximation result of $\mathcal{F}_{\text{GNN}}^W$.

**Theorem E.1.** *Let $X \subset \mathcal{G}_{m,n} \times \mathcal{H}_m^V \times \mathcal{H}_n^W$ be a compact subset that is closed under the action of $S_m \times S_n$. Suppose that $\Phi \in \mathcal{C}(X, \mathbb{R}^n)$ satisfies the followings:*

(i) *For any $\sigma_V \in S_m$, $\sigma_W \in S_n$, and $(G, H) \in X$, it holds that*

$$\Phi((\sigma_V, \sigma_W) * (G, H)) = \sigma_W(\Phi(G, H)). \tag{E.1}$$

(ii) *$\Phi(G, H) = \Phi(\hat{G}, \hat{H})$ holds for all $(G, H), (\hat{G}, \hat{H}) \in X$ with $(G, H) \overset{W}{\sim} (\hat{G}, \hat{H})$.*

(iii) *Given any $(G, H) \in X$ and any $j, j' \in \{1, 2, \ldots, n\}$, if $C_j^{l,W} = C_{j'}^{l,W}$ (the vertex colors obtained in the $l$-th iteration in WL test) holds for any $l \in \mathbb{N}$ and any choices of hash functions, then $\Phi(G, H)_j = \Phi(G, H)_{j'}$.*

*Then for any $\epsilon > 0$, there exists $F \in \mathcal{F}_{GNN}^W$ such that*

$$\sup_{(G,H) \in X} \|\Phi(G,H) - F(G,H)\| < \epsilon.$$

Theorem E.1 is a LP-graph version of results on the closure of equivariant GNN class in Azizian & Lelarge (2021); Geerts & Reutter (2022). The main tool in the proof of Theorem E.1 is the following generalized Stone-Weierstrass theorem for equivariant functions established in Azizian & Lelarge (2021).

**Theorem E.2** (Generalized Stone-Weierstrass theorem (Azizian & Lelarge, 2021, Theorem 22))**.** *Let $X$ be a compact topology space and let $\mathbf{G}$ be a finite group that acts continuously on $X$ and $\mathbb{R}^n$. Define the collection of all equivariant continuous functions from $X$ to $\mathbb{R}^n$ as follows:*

$$\mathcal{C}_E(X, \mathbb{R}^n) = \{F \in \mathcal{C}(X, \mathbb{R}^n) : F(g * x) = g * F(x), \, \forall \, x \in X, g \in \mathbf{G}\}.$$

*Consider any $\mathcal{F} \subset \mathcal{C}_E(X, \mathbb{R}^n)$ and any $\Phi \in \mathcal{C}_E(X, \mathbb{R}^n)$. Suppose the following conditions hold:*

  *(i)* $\mathcal{F}$ *is a subalgebra of* $\mathcal{C}(X, \mathbb{R}^n)$ *and* $\mathbf{1} \in \mathcal{F}$.

 *(ii) For any $x, x' \in X$, if $f(x) = f(x')$ holds for any $f \in \mathcal{C}(X, \mathbb{R})$ with $f\mathbf{1} \in \mathcal{F}$, then for any $F \in \mathcal{F}$, there exists $g \in \mathbf{G}$ such that $F(x) = g * F(x')$.*

*(iii) For any $x, x' \in X$, if $F(x) = F(x')$ holds for any $F \in \mathcal{F}$, then $\Phi(x) = \Phi(x')$.*

*(iv) For any $x \in X$, it holds that $\Phi(x)_j = \Phi(x)_{j'}, \, \forall \, (j, j') \in J(x)$, where $J(x) = \{\{1, 2, \ldots, n\}^n : F(x)_j = F(x)_{j'}, \, \forall \, F \in \mathcal{F}\}$.*

*Then for any $\epsilon > 0$, there exists $F \in \mathcal{F}$ such that*

$$\sup_{x \in X} \|\Phi(x) - F(x)\| < \epsilon.$$

We refer to Timofte (2005) for different versions of Stone-Weierstrass theorem, that is also used in Azizian & Lelarge (2021). In the proof of Theorem E.1, we also need the following lemma whose proof is almost the same as the proof of Lemma D.2 and is hence omitted.

**Lemma E.3.** $\mathcal{F}_{GNN}^W$ *is a subalgebra of* $\mathcal{C}(\mathcal{G}_{m,n} \times \mathcal{H}_m^V \times \mathcal{H}_n^W, \mathbb{R}^n)$.

*Proof of Theorem E.1.* Let $S_m \times S_n$ act on $\mathbb{R}^n$ via

$$(\sigma_V, \sigma_W) * y = \sigma_W(y), \quad \forall \, \sigma_V \in S_m, \sigma_W \in S_n, y \in \mathbb{R}^n.$$

Then it follows from (E.1) that $\Phi$ is equivariant. In addition, the definition of graph neural networks directly guarantees the equivariance of functions in $\mathcal{F}_{GNN}^W$. Therefore, one only needs to verify the conditions with $\mathcal{F}$ as $\mathcal{F}_{GNN}^W$ and $\mathbf{G}$ as $S_n$ in Theorem E.2:

• Condition (i) in Theorem E.2 follows from Lemma E.3 and the definition of $\mathcal{F}_{GNN}^W$.

• Condition (ii) in Theorem E.2 follows from Theorem 4.2 and $\mathcal{F}_{GNN}\mathbf{1} \subset \mathcal{F}_{GNN}^W$.

• Condition (iii) in Theorem E.2 follows from Corollary C.4 and Condition (ii) in Theorem E.1.

• Condition (iv) in Theorem E.2 follows from Corollary C.5 and Condition (iii) in Theorem E.1.

It finishes the proof. $\qquad\square$

## F    PROOF OF MAIN THEOREMS

We collect the proofs of main theorems stated in Section 3 in this section.

*Proof of Theorem 3.1.* If $F(G, H) = F(\hat{G}, \hat{H})$ holds for any $F \in \mathcal{F}_{\text{GNN}}$, then Theorem 4.2 guarantees that $(G, H) \sim (\hat{G}, \hat{H})$. Thus, Condition (i), (ii), and (iii) follow directly from Lemmas B.2, B.3, and B.4, respectively.

Furthermore, if $F_W(G, H) = F_W(\hat{G}, \hat{H})$, $\forall F_W \in \mathcal{F}_{\text{GNN}}^W$, then it follows from Corollary C.4 and Corollary B.7 that the two LP problems associated to $(G, H)$ and $(\hat{G}, \hat{H})$ share the same optimal solution with the smallest $\ell_2$-norm. $\square$

Then we head into the proof of three main approximation theorems, say Theorem 3.2, 3.4, and 3.6, that state that GNNs can approximate the feasibility mapping $\Phi_{\text{feas}}$, the optimal objective value mapping $\Phi_{\text{obj}}$, and the optimal solution mapping $\Phi_{\text{solu}}$ with arbitrarily small error, respectively. We have established in Sections D and E several theorems for graph neural networks to approximate continuous mappings. Therefore, the proof of Theorem 3.2, 3.4, and 3.6 basically consists of two steps:

(i) Show that the mappings $\Phi_{\text{feas}}$, $\Phi_{\text{obj}}$, and $\Phi_{\text{solu}}$ are measurable.

(ii) Use continuous mappings to approximate the target measurable mappings, and then apply the universal approximation results established in Sections D and E.

Let us first prove the feasibility of $\Phi_{\text{feas}}$, $\Phi_{\text{obj}}$, and $\Phi_{\text{solu}}$ in the following three lemmas.

**Lemma F.1.** *The feasibility mapping $\Phi_{feas}$ defined in (2.7) is measurable, i.e., the preimages $\Phi_{feas}^{-1}(0)$ and $\Phi_{feas}^{-1}(1)$ are both measurable subsets of $\mathcal{G}_{m,n} \times \mathcal{H}_m^V \times \mathcal{H}_n^W$.*

*Proof of Lemma F.1.* It suffices to prove that for any $\circ \in \{\leq, =\geq\}^m$ and any $N_l, N_u \subset \{1, 2, \ldots, n\}$, the set

$$X_{\text{feas}} := \{(A, b, l, u) \in \mathbb{R}^{m \times n} \times \mathbb{R}^m \times \mathbb{R}^{|N_l|} \times \mathbb{R}^{|N_u|} :$$
$$\exists \, x \in \mathbb{R}^n, \text{ s.t. } Ax \circ b, \; x_j \geq l_j, \; \forall \, j \in N_l, \; x_j \leq u_j, \; \forall \, j \in N_u\},$$

is a measurable subset in $\mathbb{R}^{m \times n} \times \mathbb{R}^m \times \mathbb{R}^{|N_l|} \times \mathbb{R}^{|N_u|}$. Without loss of generality, we assume that $\circ = (\leq, \ldots, \leq, =, \ldots, =, \geq, \ldots, \geq)$ where "$\leq$", "$=$", and "$\geq$" appear for $k_1$, $k_2 - k_1$, and $m - k_1 - k_2$ times, respectively, $0 \leq k_1 \leq k_2 \leq m$.

Let us define a function $V_{\text{feas}} : \mathbb{R}^{m \times n} \times \mathbb{R}^m \times \mathbb{R}^{|N_l|} \times \mathbb{R}^{|N_u|} \times \mathbb{R}^n \to \mathbb{R}_{\geq 0}$ that measures to what extend a point in $\mathbb{R}^n$ violates the constraints:

$$V_{\text{feas}}(A, b, l, u, x) = \max \left\{ \max_{1 \leq i \leq k_1} \left( \sum_{j=1}^n A_{i,j} x_j - b_i \right)_+, \max_{k_1 < i \leq k_2} \left| \sum_{j=1}^n A_{i,j} x_j - b_i \right|, \right.$$
$$\left. \max_{k_2 < i \leq m} \left( b_i - \sum_{j=1}^n A_{i,j} x_j \right)_+, \max_{j \in N_l} (l_j - x_j)_+, \max_{j \in N_u} (x_j - u_j)_+ \right\},$$

where $y_+ = \max\{y, 0\}$. It can be seen that $V_{\text{feas}}$ is continuous and $V_{\text{feas}}(A, b, l, u, x) = 0$ if and only if $Ax \circ b$, $x_j \geq l_j$, $\forall \, j \in N_l$, and $x_j \leq u_j$, $\forall \, j \in N_u$. Therefore, for any $(A, b, l, u) \in \mathbb{R}^{m \times n} \times \mathbb{R}^m \times \mathbb{R}^{|N_l|} \times \mathbb{R}^{|N_u|}$, the followings are equivalent:

- $(A, b, l, u) \in X_{\text{feas}}$.

- There exists $R \in \mathbb{N}_+$ and $x \in B_R := \{x' \in \mathbb{R}^n : \|x'\| \leq R\}$, such that $V_{\text{feas}}(A, b, l, u, x) = 0$.

- There exists $R \in \mathbb{N}_+$ such that for any $r \in \mathbb{N}_+$, $V_{\text{feas}}(A, b, l, u, x) \leq 1/r$ holds for some $x \in B_R \cap \mathbb{Q}^n$.

This implies that $X_{\text{feas}}$ can be described via

$$\bigcup_{R \in \mathbb{N}_+} \bigcap_{r \in \mathbb{N}_+} \bigcup_{x \in B_R \cap \mathbb{Q}^n} \left\{ (A, b, l, u) \in \mathbb{R}^{m \times n} \times \mathbb{R}^m \times \mathbb{R}^{|N_l|} \times \mathbb{R}^{|N_u|} : V_{\text{feas}}(A, b, l, u, x) \leq \frac{1}{r} \right\}.$$

Since $\mathbb{Q}^n$ is countable and $V$ is continuous, we immediately obtain from the above expression that $X_{\text{feas}}$ is measurable. $\qquad \square$

**Lemma F.2.** *The optimal objective value mapping $\Phi_{obj}$ defined in (2.8) is measurable.*

*Proof of Lemma F.2.* It suffices to prove that for any $\circ \in \{\leq, =\geq\}^m$, any $N_l, N_u \subset \{1, 2, \ldots, n\}$, and any $\phi \in \mathbb{R}$, the set

$$X_{\text{obj}} := \{(A, b, c, l, u) \in \mathbb{R}^{m \times n} \times \mathbb{R}^m \times \mathbb{R}^n \times \mathbb{R}^{|N_l|} \times \mathbb{R}^{|N_u|} :$$
$$\exists\, x \in \mathbb{R}^n, \text{ s.t. } c^\top x \leq \phi,\ Ax \circ b,\ x_j \geq l_j,\ \forall\, j \in N_l,\ x_j \leq u_j,\ \forall\, j \in N_u\},$$

is a measurable subset in $\mathbb{R}^{m \times n} \times \mathbb{R}^m \times \mathbb{R}^n \times \mathbb{R}^{|N_l|} \times \mathbb{R}^{|N_u|}$. Then the proof follows the same lines as in the proof of Lemma F.1, with a different violation function

$$V_{\text{obj}}(A, b, c, l, u, x) = \max\left\{(c^\top x - \phi)_+, V_{\text{feas}}(A, b, l, u, x)\right\}.$$
$\qquad \square$

**Lemma F.3.** *The optimal solution mapping $\Phi_{solu}$ defined in (2.9) is measurable.*

*Proof of Lemma F.3.* It suffices to show that for every $j_0 \in \{1, 2, \ldots, n\}$, the mapping

$$\pi_{j_0} \circ \Phi_{\text{solu}} : \Phi_{\text{obj}}^{-1}(\mathbb{R}) \to \mathbb{R},$$

is measurable, where $\pi_{j_0} : \mathbb{R}^n \to \mathbb{R}$ maps a vector $x \in \mathbb{R}^n$ to its $j_0$-th component. Similar as before, one can consider any $\circ \in \{\leq, =\geq\}^m$, any $N_l, N_u \subset \{1, 2, \ldots, n\}$, and any $\phi \in \mathbb{R}$, and prove that the set

$$X_{\text{solu}} := \{(A, b, c, l, u) \in \mathbb{R}^{m \times n} \times \mathbb{R}^m \times \mathbb{R}^n \times \mathbb{R}^{|N_l|} \times \mathbb{R}^{|N_u|} :$$
$$\text{The LP problem, } \min_{x \in \mathbb{R}^d} c^\top x, \text{ s.t. } Ax \circ b,\ x_j \geq l_j,\ \forall\, j \in N_l,\ x_j \leq u_j,\ \forall\, j \in N_u,$$
$$\text{has a finite optimal objective value, and its optimal solution with the smallest } \ell_2 - \text{norm,}$$
$$x_{\text{opt}}, \text{ satisfies } (x_{\text{opt}})_{j_0} < \phi\},$$

is measurable.

Note that we have fixed $\circ \in \{\leq, =\geq\}^m$ and $N_l, N_u \subset \{1, 2, \ldots, n\}$. Let

$$\iota : \mathbb{R}^{m \times n} \times \mathbb{R}^m \times \mathbb{R}^n \times \mathbb{R}^{|N_l|} \times \mathbb{R}^{|N_u|} \to \mathcal{G}_{m,n} \times \mathcal{H}_m^V \times \mathcal{H}_n^W,$$

be the embedding map. Define another violation function

$$V_{\text{solu}} : (\Phi_{\text{obj}} \circ \iota)^{-1}(\mathbb{R}) \times \mathbb{R}^n \to \mathbb{R},$$

via

$$V_{\text{solu}}(A, b, c, l, u, x) = \max\left\{\left(c^\top x - \Phi_{\text{obj}}(\iota(A, b, c, l, u))\right)_+, V_{\text{feas}}(A, b, c, l, u, x)\right\},$$

which is measurable with respect to $(A, b, c, l, u)$ for any fixed $x \in \mathbb{R}^n$, due to the measurability of $\Phi_{\text{obj}}$ and the continuity of $V_{\text{feas}}$. Moreover, $V_{\text{solu}}$ is continuous with respect to $x$. Therefore, the followings are equivalent for $(A, b, c, l, u) \in (\Phi_{\text{obj}} \circ \iota)^{-1}(\mathbb{R})$:

- $(A, b, c, l, u) \in X_{\text{solu}}$.

- There exists $x \in \mathbb{R}^n$ with $x_{j_0} < \phi$, such that $V_{\text{solu}}(A, b, c, l, u, x) = 0$ and $V_{\text{solu}}(A, b, c, l, u, x') > 0$, $\forall\, x' \in B_{\|x\|}$, $x'_{j_0} \geq \phi$.

- There exists $R \in \mathbb{Q}_+, r \in \mathbb{N}_+$, and $x \in B_R$ with $x_{j_0} \leq \phi - 1/r$, such that $V_{\text{solu}}(A, b, c, l, u, x) = 0$ and $V_{\text{solu}}(A, b, c, l, u, x') > 0$, $\forall\, x' \in B_R$, $x'_{j_0} \geq \phi$.

- There exists $R \in \mathbb{Q}_+$ and $r \in \mathbb{N}_+$, such that for all $r' \in \mathbb{N}_+$, $\exists\, x \in B_R \cap \mathbb{Q}^n$, $x_{j_0} \leq \phi - 1/r$, s.t. $V_{\text{solu}}(A, b, c, l, u, x) < 1/r'$ and that $\exists\, r'' \in \mathbb{N}_+$, s.t., $V_{\text{solu}}(A, b, c, l, u, x') \geq 1/r''$, $\forall\, x' \in B_R \cap \mathbb{Q}^n$, $x'_{j_0} \geq \phi$.

Therefore, one has that

$$X_{\text{solu}} = \bigcup_{R \in \mathbb{Q}_+} \bigcup_{r \in \mathbb{N}_+}$$

$$\left( \left( \bigcap_{r' \in \mathbb{N}_+} \bigcup_{x \in B_R \cap \mathbb{Q}^n, \, x_{j_0} \leq \phi - \frac{1}{r}} \left\{ (A, b, c, l, u) \in (\Phi_{\text{obj}} \circ \iota)^{-1}(\mathbb{R}) : V_{\text{solu}}(A, b, c, l, u, x) < \frac{1}{r'} \right\} \right) \right.$$

$$\left. \cap \left( \bigcup_{r'' \in \mathbb{N}_+} \bigcap_{x' \in B_R \cap \mathbb{Q}^n, \, x'_{j_0} \geq \phi} \left\{ (A, b, c, l, u) \in (\Phi_{\text{obj}} \circ \iota)^{-1}(\mathbb{R}) : V_{\text{solu}}(A, b, c, l, u, x') \geq \frac{1}{r''} \right\} \right) \right),$$

which is measurable. $\qquad\square$

With the measurability of $\Phi_{\text{feas}}$, $\Phi_{\text{obj}}$, and $\Phi_{\text{solu}}$ established, the next step is to approximate $\Phi_{\text{feas}}$, $\Phi_{\text{obj}}$, and $\Phi_{\text{solu}}$ using continuous mappings, and hence graph neural networks. Before proceeding, let us mention that $\mathcal{G}_{m,n} \times \mathcal{H}_m^V \times \mathcal{H}_n^W$ is essentially the disjoint union of finitely many product spaces of Euclidean spaces and discrete spaces that have finitely many points and are equipped with discrete measures. More specifically,

$$\mathcal{G}_{m,n} \times \mathcal{H}_m^V \times \mathcal{H}_n^W \cong \mathbb{R}^{m \times n} \times (\mathbb{R} \times \{\leq, =, \geq\})^m \times (\mathbb{R} \times (\mathbb{R} \cup \{-\infty\}) \times (\mathbb{R} \cup \{+\infty\}))^n$$

$$\cong \bigcup_{k,k'=0}^{n} \bigcup_{s=1}^{\binom{n}{k}} \bigcup_{s'=1}^{\binom{n}{k'}} \mathbb{R}^{m \times n} \times \mathbb{R}^m \times \mathbb{R}^n \times \mathbb{R}^{n-k} \times \mathbb{R}^{n-k'}$$

$$\times \{\leq, =, \geq\}^m \times \{-\infty\}^k \times \{+\infty\}^{k'}.$$

Therefore, many results in real analysis for Euclidean spaces still apply for $\mathcal{G}_{m,n} \times \mathcal{H}_m^V \times \mathcal{H}_n^W$ and $\text{Meas}(\cdot)$, including the following Lusin's theorem.

**Theorem F.4** (Lusin's theorem (Evans & Garzepy, 2018, Theorem 1.14)). *Let $\mu$ be a Borel regular measure on $\mathbb{R}^n$ and let $f : \mathbb{R}^n \to \mathbb{R}^m$ be $\mu$-measurable. Then for any $\mu$-measurable $X \subset \mathbb{R}^n$ with $\mu(X) < \infty$ and any $\epsilon > 0$, there exists a compact set $E \subset X$ with $\mu(X \backslash E) < \epsilon$, such that $f|_E$ is continuous.*

*Proof of Theorem 3.2.* Since $X \subset \mathcal{G}_{m,n} \times \mathcal{H}_m^V \times \mathcal{H}_n^W$ is measurable with finite measure, according to Lusin's theorem, there is a compact set $E \subset X$ such that $\Phi_{\text{feas}}|_E$ is continuous with $\text{Meas}(X \backslash E) < \epsilon$. By Lemma B.2 and Theorem 4.3, there exists $F \in \mathcal{F}_{\text{GNN}}$ such that

$$\sup_{(G,H) \in E} |F(G, H) - \Phi_{\text{feas}}(G, H)| < \frac{1}{2},$$

which implies that

$$\mathbb{I}_{F(G,H) > 1/2} = \Phi_{\text{feas}}(G, H), \quad \forall \, (G, H) \in E.$$

Therefore, one obtains

$$\text{Meas}\left(\left\{ (G, H) \in X : \mathbb{I}_{F(G,H) > 1/2} \neq \Phi_{\text{feas}}(G, H) \right\}\right) \leq \text{Meas}(X \backslash E) < \epsilon,$$

and the proof is completed. $\qquad\square$

*Proof of Corollary 3.3.* As a finite set, $\mathcal{D}$ is compact and $\Phi_{\text{feas}}|_{\mathcal{D}}$ is continuous. The rest of the proof is similar to that of Theorem 3.2, using Lemma B.2 and Theorem 4.3. $\qquad\square$

*Proof of Theorem 3.4.* (i) The proof follows the same lines as in the proof of Theorem 3.2, with the difference that we approximate

$$\Phi(G, H) = \begin{cases} 1, & \text{if } \Phi_{\text{obj}}(G, H) \in \mathbb{R}, \\ 0, & \text{otherwise}, \end{cases}$$

instead of $\Phi_{\text{feas}}$ and we use Lemma B.3 instead of Lemma B.2.

(ii) The proof is still similar to that of Theorem 3.2. By Lusin's theorem, there is a compact set $E \subset X \cap \Phi_{\text{obj}}^{-1}(\mathbb{R})$ such that $\Phi_{\text{obj}}|_E$ is continuous with $\text{Meas}\left((X \cap \Phi_{\text{obj}}^{-1}(\mathbb{R}))\backslash E\right) < \epsilon$. Using Lemma B.3 and Theorem 4.3, there exists $F_2 \in \mathcal{F}_{\text{GNN}}$ with

$$\sup_{(G,H) \in E} |F_2(G,H) - \Phi_{\text{obj}}(G,H)| < \delta,$$

which implies that

$$\text{Meas}\left(\{(G,H) \in X : |F_2(G,H) - \Phi_{\text{obj}}(G,H)| > \delta\}\right) \leq \text{Meas}\left((X \cap \Phi_{\text{obj}}^{-1}(\mathbb{R}))\backslash E\right) < \epsilon.$$

$\square$

*Proof of Corollary 3.5.* The results can be proved using similar techniques as in Theorem 3.2 and Theorem 3.4 by noticing that any finite dataset is compact on which any real-valued function is continuous. $\square$

*Proof of Theorem 3.6.* Without loss of generality, we can assume that $X \subset \Phi_{\text{obj}}^{-1}(\mathbb{R}) \subset \mathcal{G}_{m,n} \times \mathcal{H}_m^V \times \mathcal{H}_n^W$ is closed under the action of $S_m \times S_n$; otherwise, we can replace $X$ by $\bigcup_{(\sigma_V, \sigma_W) \in S_m \times S_n} (\sigma_V, \sigma_W) * X$. By Lusin's theorem, there exists a compact subset $E' \subset X$ such that $\text{Meas}(A\backslash X) < \epsilon/|S_m \times S_n|$ and that $\Phi_{\text{solu}}|_{E'}$ is continuous. Define another compact set:

$$E = \bigcap_{(\sigma_V, \sigma_W) \in S_m \times S_n} (\sigma_V, \sigma_W) * E' \subset X,$$

which is closed under the action of $S_m \times S_n$ and satisfies

$$\begin{aligned}
\text{Meas}(X\backslash E) &\leq \sum_{(\sigma_V, \sigma_W) \in S_m \times S_n} \text{Meas}(X\backslash(\sigma_V, \sigma_W) * E') \\
&= \sum_{(\sigma_V, \sigma_W) \in S_m \times S_n} \text{Meas}((\sigma_V, \sigma_W) * X \backslash (\sigma_V, \sigma_W) * E') \\
&= \sum_{(\sigma_V, \sigma_W) \in S_m \times S_n} \mu((\sigma_V, \sigma_W) * (X\backslash E')) \\
&= \sum_{(\sigma_V, \sigma_W) \in S_m \times S_n} \mu(X\backslash E') \\
&< \sum_{(\sigma_V, \sigma_W) \in S_m \times S_n} \frac{\epsilon}{|S_m \times S_n|} \\
&= \epsilon.
\end{aligned}$$

Note that the three conditions in Theorem E.1 are satisfied by the definition of $\Phi_{\text{solu}}$, Corollary B.7, and Corollary B.6, respectively. Using Theorem E.1, there exists $F_W \in \mathcal{F}_{\text{GNN}}^W$ such that

$$\sup_{(G,H) \in E} \|F_W(G,H) - \Phi_{\text{solu}}(G,H)\| < \delta.$$

Therefore, it holds that

$$\text{Meas}\left(\{(G,H) \in X : \|F_W(G,H) - \Phi_{\text{solu}}(G,H)\| > \delta\}\right) \leq \text{Meas}(X\backslash E) < \epsilon,$$

which completes the proof. $\square$

*Proof of Corollary 3.7.* One can assume that $\mathcal{D}$ is closed under the action of $S_m \times S_n$; otherwise, a larger but still finite dataset, $\bigcup_{(\sigma_V, \sigma_W) \in S_m \times S_n} (\sigma_V, \sigma_W) * \mathcal{D}$, can be considered instead of $\mathcal{D}$. The rest of the proof is similar to that of Theorem 3.6 since $\mathcal{D}$ is compact and $\Phi_{\text{solu}}|_{\mathcal{D}}$ is continuous. $\square$

# G  DETAILS OF THE NUMERICAL EXPERIMENTS AND EXTRA EXPERIMENTS

**LP instance generation**  We generate each LP with the following way. We set $m = 10$ and $n = 50$. Each matrix $A$ is sparse with 100 nonzero elements whose positions are sampled uniformly and values are sampled normally. Each element in $b, c$ are sampled i.i.d and uniformly from $[-1, 1]$. Additionally, each element in $c$ is scaled by $0.01$. The variable bounds $l, u$ are sampled with $\mathcal{N}(0, 10)$. If $l_j > u_j$, then we swap $l_j$ and $u_j$ for all $1 \leq j \leq n$. Furthermore, we sample $\circ_i$ i.i.d with $\mathbb{P}(\circ_i = \text{``} \leq \text{''}) = 0.7$ and $\mathbb{P}(\circ_i = \text{``} = \text{''}) = 0.3$. With the generation approach above, the probability that each LP to be feasible is around $0.53$.

**MLP architectures**  As we mentions in the main text, all the learnable functions in GNN are taken as MLPs. The input functions $f_{\text{in}}^V, f_{\text{in}}^W$ have one hidden layer and other functions $f_{\text{out}}, f_{\text{out}}^W, \{f_l^V, f_l^W, g_l^V, g_l^W\}_{l=0}^L$ have two hidden layers. The embedding size $d_0, \cdots, d_L$ are uniformly taken as $d$ that is chosen from $\{2, 4, 8, 16, 32, 64, 128, 256, 512\}$. All the activation functions are ReLU.

**Training settings**  We use Adam (Kingma & Ba, 2014) as our training optimizer with learning rate of 0.0003. The loss function is taken as mean squared error. All the experiments are conducted on a Linux server with an Intel Xeon Platinum 8163 GPU and eight NVIDIA Tesla V100 GPUs.

Extra experiments on generalization We generate the testing set consisting of 1000 LP problems using the same distribution as that of the training set. The performance of the trained GNNs on training set and testing set is presented in Table 1, 2, and 3 for feasibility, optimal objective value, and optimal solution, respectively. The metric in Table 1 is the rate of classification errors; the metric in Table 2 is a relative error defined as $|F - \Phi_{\text{obj}}|/(|\Phi_{\text{obj}}| + 1)$; the metric in Table 3 is defined with $\|F_W - \Phi_{\text{solu}}\|/(\|\Phi_{\text{solu}}\| + 1)$.

| Number of Training Samples | 100 | 500 | 2500 |
|---|---|---|---|
| The Error on the training set | 0 | 0 | 0.067 |
| The Error on the testing set | 0.454 | 0.339 | 0.175 |

Table 1: Generalization for feasibility (Num. GNN parameters: 1254)

| Number of Training Samples | 100 | 500 | 2500 |
|---|---|---|---|
| The Error on the training set | 1.9e-6 | 0.080 | 0.128 |
| The Error on the testing set | 0.790 | 0.591 | 0.173 |

Table 2: Generalization for optimal objective value (Num. GNN parameters: 1254)

| Number of Training Samples | 100 | 500 | 2500 |
|---|---|---|---|
| The Error on the training set | 0.141 | 0.193 | 0.205 |
| The Error on the testing set | 0.550 | 0.351 | 0.274 |

Table 3: Generalization for optimal solution (Num. GNN parameters: 7888)

One can observe that, for a GNN with fixed size, its generalization performance, i.e., the performance on the testing set is increasing if it is trained with more training samples. Given these numerical results, we believe that understanding the generalization quantitatively and theoretically deserves future research.

