# OpenReview forum: "On Representing Linear Programs by Graph Neural Networks"
_ICLR.cc/2023/Conference — ICLR 2023 notable top 25%_

### Official Review · Reviewer_ubzC · 2022-10-24

**Confidence:** 3
**Correctness:** 4
**Technical Novelty And Significance:** 4
**Empirical Novelty And Significance:** Not applicable
**Recommendation:** 8

**Clarity, Quality, Novelty And Reproducibility:**

The authors claim that "we established the first complete proof that GNN can universally represent a broad class of LPs".
I am not expert enough to be confident about this, but if so then this looks like a valuable foundational result.
Furthermore the argument based on the WL test is conceptually very simple and relates to much previous work on GNNs.


**Strength And Weaknesses:**

Strengths: what looks like a general and solid foundational theorem is clearly presented, with a conceptually simple argument.

Weaknesses: the results do not bound the size of the networks required and do not cover generalization, so are of only theoretical interest.

**Summary Of The Paper:**

This paper studies the ability of GNNs to solve linear programming problems.  It is quite systematic and proves that within a well defined class of GNNs, and given a set of LPs, there exist networks (with specified weights) which can distinguish pairs of LP problems within the set with different feasibility and objective values, and which can approximate the objective value.  The argument is that the Weisfeiler-Lehman test, which it is known that GNNs can perform, suffices for these tasks.  A simple experimental check is made.

**Summary Of The Review:**

A very interesting theoretical contribution to study of GNN.

---

> ### Author Response · Authors · 2022-11-13
> **Reply to Reviewer ubzC**
>
> Thank you very much for your encouraging comments! Although our work does not consider the size/complexity of GNNs or the generalization, we would like to add some discussions in the last paragraph of the main text and provide some related experimental results in the appendix.
>
> **Added discussions.**
> Our results illustrate the possibility of representing LP characters with GNNs, while the size or complexity of GNNs is not guaranteed, which will be a significant future direction. Another interesting topic is the generalization: the number of training samples needed to train a good enough GNN. We provide some generalization results in the appendix.
>
> **Extra experiments in the appendix.**
> To test the generalization performance of our trained GNNs, we generate extra 1000 LP instances with the same distribution of the training set. The experiment results are reported in the following table. Our results show that, with a fixed model size, the testing error reduces as the number of training samples increases until it matches the training error. To obtain better generalization performance, one should collect more training samples. The number of training samples and its relatinship with the models' size would be an interesting future topic.
>
> Feasibility (Num. GNN parameters: 1254):
> | Num. Training Samps. | 100   | 500   | 2500  |
> |----------------------|-------|-------|-------|
> | Training Error       | 0     | 0     | 0.067 |
> | Testing Error        | 0.454 | 0.339 | 0.175 |
>
> Objective (Num. GNN parameters: 1254):
> | Num. Training Samps. | 100   | 500   | 2500  |
> |----------------------|-------|-------|-------|
> | Training Error       | 1.9e-6| 0.080   | 0.128 |
> | Testing Error        | 0.790 | 0.591 | 0.173 |
>
> Solution (Num. GNN parameters: 7888):
> | Num. Training Samps. | 100   | 500   | 2500  |
> |----------------------|-------|-------|-------|
> | Training Error       | 0.141    | 0.193    | 0.205 |
> | Testing Error        | 0.550 | 0.351 | 0.274 |
>
>
> Finally, we make some clarifications to fix your concerns on **novelty.** Although representing generic LPs or MILPs with GNNs has been proposed in the literature since 2019, there are still no theoretical foundations for such approach, to the best of our knowledge. Maybe the most related papers are [1,2] that have been cited in our paper. Both [1] and [2] studied **specific** graph-related (optimization) problems. [1] studied the minimum dominating set problem, maximum matching problem, and minimum vertex cover problem; [2] studied cycle detection, subgraph verification, minimum spanning tree, minimum cut, and diameter computation. While our paper shows results on **generic** LP problem, which is different from [1,2].
>
> [1] Sato, Ryoma, Makoto Yamada, and Hisashi Kashima. "Approximation ratios of graph neural networks for combinatorial problems." Advances in Neural Information Processing Systems 32 (2019).
>
> [2] Loukas, Andreas. "What graph neural networks cannot learn: depth vs width." International Conference on Learning Representations 2020.

---

### Official Review · Reviewer_AwvS · 2022-10-25

**Confidence:** 3
**Correctness:** 4
**Technical Novelty And Significance:** 3
**Empirical Novelty And Significance:** Not applicable
**Recommendation:** 6

**Clarity, Quality, Novelty And Reproducibility:**

The paper is well written and easy to follow. The examples given, such as the LP-Graph in Figure 1 on page 3 and the LP-Graphs indistinguishable by WL test in Figure 2 on page 8, are very illustrative.

In general, both the big picture and the technical details are explained well, complete with experimental validation.

In terms of novelty, the idea of connecting LP graph or properties with GNN or WL test appears to be new, and mathematically natural in hindsight.

The paper presented sufficient details so the experimental results should be reproducible.

**Strength And Weaknesses:**

Strengths:
The solution given to the problem is natural in hindsight, and the connection between LP graphs/properties and GNN/WL-test appear to be new. And GNN and LP are both important on their own.

Weaknesses:
The task of learning different LP properties (feasibility, optimal value, and shortest optimal solution) by neural networks appears somewhat artificial, because there are much better tools out there (various LP solvers, including those used by this paper in the numerical experiments section) than neural networks for this task, and likely neural networks will never be remotely competitive (both in theory or in practice) for learning LP related properties, where a deep mathematical theory trumps neural networks and vast training data.

**Summary Of The Paper:**

This paper studies the problem of learning Linear Program (LP) properties—feasibility, optimal value, and shortest optimal solution—by Graph Neural Networks (GNNs). By extending existing results on GNNs—namely the paper by Xu et al. 2019 which equates the separation power of GNNs with Weisfeiler–Lehman isomorphism test (WL test), and the Generalized Stone-Weierstrass Theorem as in Azizian &
Lelarge 2021—this paper shows that GNNs can separate and represent LP properties.


**Summary Of The Review:**

This paper could be an interesting, if slightly incremental, contribution to the mathematical theory of GNN or WL tests.

---

> ### Author Response · Authors · 2022-11-13
> **Reply to Reviewer AwvS (Part I)**
>
> Thank you very much for your valuable comments! Your main concern is about the practice of neural networks (NNs) in an LP solver. We agree with you that totally **replacing** an LP solver with a NN is not wise (at least for now), while **helping** LP solvers with neural networks is quite promising. For example, NNs can help an LP solver choose algorithms instance-adaptively. An LP solver is actually a collection of several related algorithms, and how to choose the most efficient algorithm for each LP instance is a significant problem.
>
> To support this point, we conduct an experiment with a state-of-the-art commercial solver CPLEX on the dataset MIPLIB2017, and, for each instance, we choose between two LP algorithms: the primal simplex method and the dual simplex method. In that experiment, we trained a NN that maps an LP instance to its optimal decision (PrimalSpx or DualSpx) and tested the model on test sets where each instance is not included in the training set. Our testing results are reported in following tables. With the NN-suggested adaptive configuration, CPLEX obtained clear performance improvement compared with either PrimalSpx or DualSpx. Details can be found at the end of this rebuttal. Of course, one may **manually** design such adaptive configuration rules instead of training a neural network; however, such a process becomes extremely painful when the size of the LP set grows. With machine learning, one may **automate** this process, and this is one of the main purposes of "learning to optimize."
>
> Geometric Mean of running time (secs):
> |             Test Set            | Set 1 | Set 2 | Set 3 | Set 4 | Set 5 |
> |:-------------------------------:|:-----:|:-----:|:-----:|:-----:|:-----:|
> |           Primal Spx.           |  7.19 | 12.46 |  8.88 | 11.95 |  7.64 |
> |            Dual Spx.            |  7.04 |  9.44 |  7.58 |  9.28 |  7.50 |
> |             Adaptive            |  6.09 |  8.68 |  6.75 |  8.94 |  6.33 |
>
> Number of LP instances with running time > 100 secs:
> |             Test Set            | Set 1 | Set 2 | Set 3 | Set 4 | Set 5 |
> |:-------------------------------:|:-----:|:-----:|:-----:|:-----:|:-----:|
> |           Primal Spx.           |   15  |   23  |   18  |   22  |   18  |
> |            Dual Spx.            |   18  |   18  |   18  |   19  |   15  |
> |             Adaptive            |   15  |   15  |   15  |   16  |   13  |
>
> Number of LP instances with running time > 10 secs:
> |             Test Set            | Set 1 | Set 2 | Set 3 | Set 4 | Set 5 |
> |:-------------------------------:|:-----:|:-----:|:-----:|:-----:|:-----:|
> |           Primal Spx.           |   34  |   44  |   42  |   49  |   42  |
> |            Dual Spx.            |   31  |   38  |   33  |   47  |   39  |
> |             Adaptive            |   28  |   34  |   30  |   44  |   34  |
>
> Given the above discussions, we will answer the question: why GNN representing the feasibility and optimal value of LP is meaningful? To suggest a proper adaptive decision/configuration for each LP instance, a neural network model should be able to **capture the key characters** of an instance. Otherwise, if a model equally treat two extremely different LP instances, one cannot expect that model suggest useful adaptive information. Consequently, feasibility and optimal value, two significant characters of an LP, are good metrics to evaluate the representation power of a machine learning model on LP problems.
>
> Furthermore, we would like to clarify that, the purpose of approximating the LP optimal solution is also not to **replace** LP solvers, but to **warmstart** LP solvers. Some very recent papers [1,2,3] on DC optimal power flow (DC-OPF), an important type of LP problems, experimentally show the possibility of fast approximating LP solutions with deep neural networks. One may initialize an LP solver with those approximated solutions.
>
> Finally, we added the above discussions in our updated draft and clarify our motivation.
>
> [1] Deka, Deepjyoti, and Sidhant Misra. (2019) "Learning for DC-OPF: Classifying active sets using neural nets."
>
> [2] Pan, Xiang, et al. (2021) "DeepOPF: deep neural networks for optimal power flow."
>
> [3] Chen, Yize, et al. (2022) “Learning to Solve DCOPF: A Duality Approach.”

---

> > ### Author Response · Authors · 2022-11-13
> > **Reply to Reviewer AwvS (Part II)**
> >
> > **Details of the experiment.** MIPLIB2017 has 1064 MILP instances, and we work on their LP relaxations. We first remove those instances that cannot be solved by either PrimalSpx or DualSpx, and the dataset has 1014 instances. Then we divide the dataset into 5 sets randomly; the sets are named Set1, ..., and Set5, respectively. We conduct cross-validation on these sets: training a NN with 4 of the 5 sets and testing the NN on the rest. For each instance in the training set, we run the CPLEX solver with both PrimalSpx and DualSpx and then make labels depending on which algorithm is better. We choose a multilayer perceptron (MLP) as the model, and its input consists of 15 hand-crafted features. The CPLEX version we used is 12.7. Finally, we report the average inference time of the trained NNs for each instance: **5.07ms**, which is really cheap compared to the running time of LP solvers.

---

### Official Review · Reviewer_XXXE · 2022-10-26

**Confidence:** 2
**Correctness:** 3
**Technical Novelty And Significance:** 2
**Empirical Novelty And Significance:** Not applicable
**Recommendation:** 5

**Clarity, Quality, Novelty And Reproducibility:**

The exposition is clear.
The paper consists of minor grammar and spelling mistakes.

**Strength And Weaknesses:**

The work is missing a clear explanation of why the WL test limit is related to the task of representing an LP's optimality.

It would be interesting to see results and comparisons between the following special cases of CSPs:
1. Linear programming in which the variables are real, and the factors are linear inequalities.
2. Integer linear programming in which the variables are integers and the factors are linear inequalities.
3. Mixed integer programming in which the variables are reals and integers and the factors are linear inequalities.

**Summary Of The Paper:**

Linear programs (LPs) are a special case of constraint satisfaction problems (CSPs).
This work claims that GNN's may approximate the properties of LPs: if it is feasible, unbounded and its optimal solution if bounded.
namely: feasibility, optimal objective value, and optimal solution.
The work shows (in theorems 4.1 and 4.2 ) that the limitations of GNNs are not a problem for demonstrating these properties.
The work validates using numerical results (in section 5) that GNNs may approximate the optimal objective value and solution (in theorem 3.4).

**Summary Of The Review:**

The presentation of the relationship to previous work on theoretical guarantees of optimality is lacking in the related work section.
The contribution is marginally significant and novel compared with previous work.

---

> ### Author Response · Authors · 2022-11-13
> **Reply to Reviewer XXXE**
>
> Thank you very much for your helpful comments! Please find our responses below.
>
> **(WL test versus LP).** First, we would like to clarify that linking GNN and mixed-integer linear programming or its LP relaxation together is not only an imagination. Some recent studies [1,2,3] show that GNNs can be incorporated into (MI)LP solvers and help (MI)LP solvers to achieve better performance. To theoretically understand the benefits and limitations of such an approach, one has to answer (at least) three related questions:
> * (Existence). Do there exist GNNs that can represent key characters of LPs? This question is also named as the representation power of GNNs. If the answer was no, GNNs would confuse an LP instance with another extremely distinct LP instance, and one could not expect GNN as a good tool for LP and even an ambitious goal, MILP. Thus, this is actually a fundamental question.
> * (Trainability). If such GNNs exist, can we find them? The process of finding such GNNs is also named as training.
> * (Generalization). Does the trained GNN work well on unseen instances? This is also named as the generalization performance of GNNs.
>
> To the best of our knowledge, the answers to the three questions are still lacking. In this paper, we focus on the first question. **We added some discussions in the introduction to further clarify our motivation.**
>
> Since the separation power of GNNs is actually equal to the WL test (Xu et al., 2019), one would expect that the limitation of the WL test might restrict GNNs from universally representing LP, and this is the motivation to link the WL test and LP together. **We rephrased Section 4 to fix this concern.**
>
> **(Contributions).** Based on the above discussions, our work shows a positive theoretical result among those negative results on the limitation of GNNs, and shows the possibility of helping LP solvers with GNNs. Given the rising interest in "machine learning for optimization," we believe that our results provide interesting insights to the community. Could you explain more on why you think "The contribution is marginally significant and novel compared with previous work"?
>
> **(Comparison of LP, ILP and MILP).** We provide a comparison result here. By slightly modifying the examples in Figure 2 in our paper, one could obtain one integer linear program and one mixed-integer linear program:
>
> $$
> \begin{aligned}
> \min_{x} & \sum_{i=1}^4 x_i, \\\\
> \text{s.t. } & x_1+x_2=1, x_2+x_3=1, x_3+x_4=1,~x_4+x_1=1, \\\\
> & x_i \geq 0, x_i \in \mathbb{Z}, \text{for all } 1\leq i \leq 4.
> \end{aligned}
> $$
>
> This instance has two solutions $(1,0,1,0)$ and $(0,1,0,1)$. It can be shown that, WL test treats all the four variables $x_1,x_2,x_3,x_4$ equally and cannot distinguish the four variables. The same conclusion holds for GNNs, and, consequently, any GNNs that yields the formula in our paper never accurately predict either of the two solutions.
>
> $$
> \begin{aligned}
> \min_{x} & \sum_{i=1}^4 x_i, \\\\
> \text{s.t. } & x_1+x_2=1,x_2+x_1=1,x_3+x_4=1,x_4+x_3=1,\\\\
> & x_i \geq 0, \text{for all } 1\leq i \leq 4, \\\\
> & x_1,x_2 \in \mathbb{Z}.
> \end{aligned}
> $$
>
> For this instance, we have either $(x_1,x_2)=(1,0)$ or $(x_1,x_2)=(0,1)$. However, with the same argument as the above instance, WL test treats $x_1$ and $x_2$ equally and never distinguish them. Thus, GNNs will not predict any of the solutions to this instance.
>
>
> **(Gurantees of optimality).** Could you provide a more **precise** definition of "theoretical guarantees of optimality"? And we would appreciate if you could provide some **specific** references. We would like to clarify that, the purpose of this paper is not to guarantee the optimality of LPs, but we want to identify whether GNNs can fast approximate key characters of LPs. Thus, we do not think that it is necessary to include "the relationship to previous work on theoretical guarantees of optimality."
>
>
> **(Minor correctness issues).** You mention that there are some minor correctness issues. We highly appreciate it if you can mention some details to help us improve the manuscript.
>
>
> [1] Li, Xijun, et al. (2022) "Learning to Reformulate for Linear Programming."
>
> [2] Nair, Vinod, et al. (2020) "Solving mixed integer programs using neural networks."
>
> [3] Gasse, Maxime, et al. (2019) "Exact combinatorial optimization with graph convolutional neural networks."

---

### Decision · Program_Chairs · 2023-01-20

**Decision:**

Accept: notable-top-25%

**Justification For Why Not Higher Score:**

Not clear that the paper has the broad applicability and interest to justify an oral presentation.

**Justification For Why Not Lower Score:**

Paper is high quality and seems like a foundational result for the area of GNNs for LPs.

**Metareview: Summary, Strengths And Weaknesses:**

In recent years there has been an emerging line of work on using graph neural networks (GNNs) to help with solving linear programs. One may wonder about the representational power of how well-suited GNNs to this task, and as the authors nicely summarize in the author response to XXXE, there are a series of questions that one could ask:

"""
(Existence). Do there exist GNNs that can represent key characters of LPs? This question is also named as the representation power of GNNs. If the answer was no, GNNs would confuse an LP instance with another extremely distinct LP instance, and one could not expect GNN as a good tool for LP and even an ambitious goal, MILP. Thus, this is actually a fundamental question.

(Trainability). If such GNNs exist, can we find them? The process of finding such GNNs is also named as training.

(Generalization). Does the trained GNN work well on unseen instances? This is also named as the generalization performance of GNNs.
"""

This paper addresses the first question by bringing together results linking GNN representational power to the Weisfeiler–Lehman (WL) isomorphism test and results linking WL to properties of LPs. The paper is well-scoped and high quality. The main weakness that I see (raised by ubzC) is that "the results do not bound the size of the networks required and do not cover generalization, so are of only theoretical interest." However, this seems appropriate to leave as a challenge for future work.



**Note From Pc:**

if the above contains the word "oral" or "spotlight" please see: "oral" presentation means -> notable-top-5% and "spotlight" means -> notable-top-25%. As stated in our emails, we are disassociating presentation type from AC recommendations